# Backbone-independent NMR resonance assignments of methyl probes in large proteins

Santrupti Nerli[1,2,4], Viviane S. De Paula [2,3,4], Andrew C. McShan[2] & Nikolaos G. Sgourakis [2✉]

Methyl-specific isotope labeling is a powerful tool to study the structure, dynamics and interactions of large proteins and protein complexes by solution-state NMR. However, widespread applications of this methodology have been limited by challenges in obtaining confident resonance assignments. Here, we present Methyl Assignments Using Satisfiability (MAUS), leveraging Nuclear Overhauser Effect cross-peak data, peak residue type classification and a known 3D structure or structural model to provide robust resonance assignments consistent with all the experimental inputs. Using data recorded for targets with known assignments in the 10–45 kDa size range, MAUS outperforms existing methods by up to 25,000 times in speed while maintaining 100% accuracy. We derive de novo assignments for multiple Cas9 nuclease domains, demonstrating that the methyl resonances of multi-domain proteins can be assigned accurately in a matter of days, while reducing biases introduced by manual pre-processing of the raw NOE data. MAUS is available through an online web-server.

[1] Department of Biomolecular Engineering, University of California, Santa Cruz, CA 95064, USA. [2] Center for Computational and Genomic Medicine, Department of Pathology and Laboratory Medicine, The Children's Hospital of Philadelphia and Department of Biochemistry and Biophysics, Perelman School of Medicine, University of Pennsylvania, 3501 Civic Center Boulevard, Philadelphia, PA 19104, USA. [3] Núcleo Multidisciplinar de Pesquisa em Biologia, Universidade Federal do Rio de Janeiro, Duque de Caxias, RJ 25245-390, Brazil. [4] These authors contributed equally: Santrupti Nerli, Viviane S. De Paula. ✉email: nikolaos.sgourakis@pennmedicine.upenn.edu

The use of methyl probes has opened new avenues for the application of nuclear magnetic resonance (NMR) methods to study large molecular machines[1]. The signal enhancement offered by methyl-transverse relaxation optimized spectroscopy (TROSY) techniques[2] and a suite of experiments for quantitative characterization of protein dynamics occurring over a broad range of timescales[3] have rendered methyl-based NMR a formidable tool for detailed mechanistic studies of important biological systems[4]. The main bottleneck in applications of methyl-based NMR is obtaining confident resonance assignments. In the conventional approach, backbone assignments are first established using triple-resonance experiments[5]. Then, methyl resonances are connected to the backbone using either methyl out-and-back experiments[6] or, more commonly, using $^{15}N$ and $^{13}C$ edited amide-to-methyl nuclear Overhauser effect (NOE) measurements[7]. However, in the absence of previously established backbone assignments, deriving confident assignments for methyl resonances remains a challenge. Although site-directed mutagenesis of individual methyl-bearing residues[8] provides unambiguous assignments, the laborious, costly, and time-consuming nature of this approach limits applications to study larger, multi-domain proteins.

Recently, several automated methyl assignment methods have been proposed[9–12]. To circumvent the need for existing backbone assignments, methyl NOE data and a known structure of the target protein can be used to derive a set of possible assignments, by fitting local NOE networks to methyl distances derived from the three-dimensional (3D) structure. In general, these methods start from fitting sub-sets of NOE connectivities to local clusters of methyls in the structure and then expand those to derive self-consistent assignments for the remaining methyl resonances. The NOE peak intensities are then used in an optimization process aiming to further reduce the space of solutions. However, any method that uses local information is bound to make mistakes globally and, as a result, most of these methods can miss the correct (ground truth) assignments from their scope of solutions, yielding a significant (up to 55%) error rate. An alternative approach is to perform an exhaustive mapping of the global NOE network to the target structure. This strategy was first outlined in the method MAGMA[13], which treats methyl assignments as a maximum subgraph matching problem, and invokes a graph theory algorithm[14] to enumerate all assignments, which satisfy the maximum number of NOE connectivities. If the input contains only true positive NOE data (i.e., the NOE data graph is a subgraph of the structure graph), then MAGMA resorts to the VF2 algorithm[15]. Alternatively, maximum subgraph matching provides a reasonable compromise to account for the minimum number of false-positive NOEs, with the caveat that the ground truth solution may entail additional misidentified NOEs in the input. Moreover, MAGMA requires users to provide an annotated data graph, derived through an extensive, manual analysis of 3D or four-dimensional (4D) NOE spectroscopy (NOESY) peak data, which introduces an additional processing step, limiting its use by non-experts.

In the present work, we describe an automated system (MAUS: Methyl Assignments Using Satisfiability), which first formulates a set of rules, and then provides a compact description of all assignment possibilities that are consistent with these rules. Specifically, MAUS generates a structure graph, $G$, representing all methyl NOE connectivities present in an input Protein Data Bank (PDB) structure or structural model of the protein of interest, and multiple independent data graphs, $H$, containing all possible NOE networks, which can be derived from a list of raw 3D or 4D NOESY peaks. The NOE network is supplemented with residue type, stereospecificity, and geminal methyl connectivity constraints. Then, MAUS leverages an efficient algorithm to determine all valid ways of mapping every $H$ into $G$ (termed subgraph isomorphism), which respects all the experimental inputs. We test our method on a benchmark set of protein targets in the 10–45 kDa size range and show that MAUS maintains a robust performance, providing 100% accurate assignments at high levels of completeness, while offering a significant performance advantage relative to existing methods using the same inputs. Using MAUS, the methyl resonances of large, multi-domain proteins can be assigned accurately in a matter of days, completely bypassing the need for more laborious backbone-based NMR spectroscopy approaches.

## Results and discussion

**Methyl assignments as a subgraph isomorphism problem.** Rather than treating the methyl assignments as a maximum subgraph matching problem[13], MAUS models the NOE data as a sparse sample of all possible connectivities present in the input structure. MAUS uses the NOE network together with additional experimental inputs, such as peak residue type information and geminal methyl resonance connectivities, to build a system of hard constraints. The constraints outline a subgraph isomorphism problem of fitting a sparse data graph $H$, into the original structure graph $G$ (see "Methods" and Fig. 1a).

To fully account for all methyl connectivities consistent with the input structure and spin diffusion effects[16], alternative side-chain rotamers are modeled using the program *Rosetta*[17] and maximum distances of 8 and 10 Å are applied to derive a structure graph, $G$, containing all possible short- and long-range NOEs, respectively (Fig. 1b). In addition, MAUS explicitly considers (i) all possible mappings of 3D NOE cross-peaks to two-dimensional (2D) reference peaks, or clusters (Fig. 1c), and (ii) all possible matchings between upper-diagonal and lower-diagonal NOE cross-peaks, formally analyzed as connected components of a bipartite symmetrization graph (Fig. 1d). This approach relieves the user from the burden of interpreting the raw data (3D or 4D NOE peaks), through an explicit and exhaustive consideration of all possible data graphs consistent with the input NOE peaks (Fig. 1e).

MAUS leverages a special-purpose constraint satisfaction solver (SAT) to enumerate all valid assignments using an iterative ansatz (see "Methods"). Using 3275 simulations of NOE data and structure graphs from a non-redundant set of 147 protein structures, we show that, relative to the VF2 algorithm[15], SAT maintains a robust performance for problems of different sizes, topologies, and data sparsity levels, delivering accurate assignments in a matter of seconds (Supplementary Fig. 1). Finally, using the MAGMA benchmark set of eight targets[13] with the identical structure and data graphs provided to both programs, we find that, although MAGMA is marginally faster than MAUS for targets, which can be solved relatively quickly by both methods, MAUS maintains a consistent performance across all targets, including larger, more complex cases such as the 81.4 kDa maltose synthase G (Supplementary Table 1, Supplementary Results).

**MAUS workflow and results on targets with known assignments.** We tested MAUS using representative data sets recorded for a benchmark set of four protein targets spanning a range of sizes, folds and domain complexities: human $\beta_2$-microglobulin (H$\beta_2$m; 12 kDa, all-$\beta$-fold, single domain), maltose-binding protein (MBP; 41 kDa, all-$\alpha$-fold, two-domain), and two major histocompatibility complex class-I (MHC-I) molecules of divergent heavy-chain sequences (HLA-A01; 45.5 kDa and HLA-A02; 44.8 kDa, mixed $\alpha/\beta$-fold, three-domain) (Table 1). The X-ray structures and ground truth assignments for these targets were

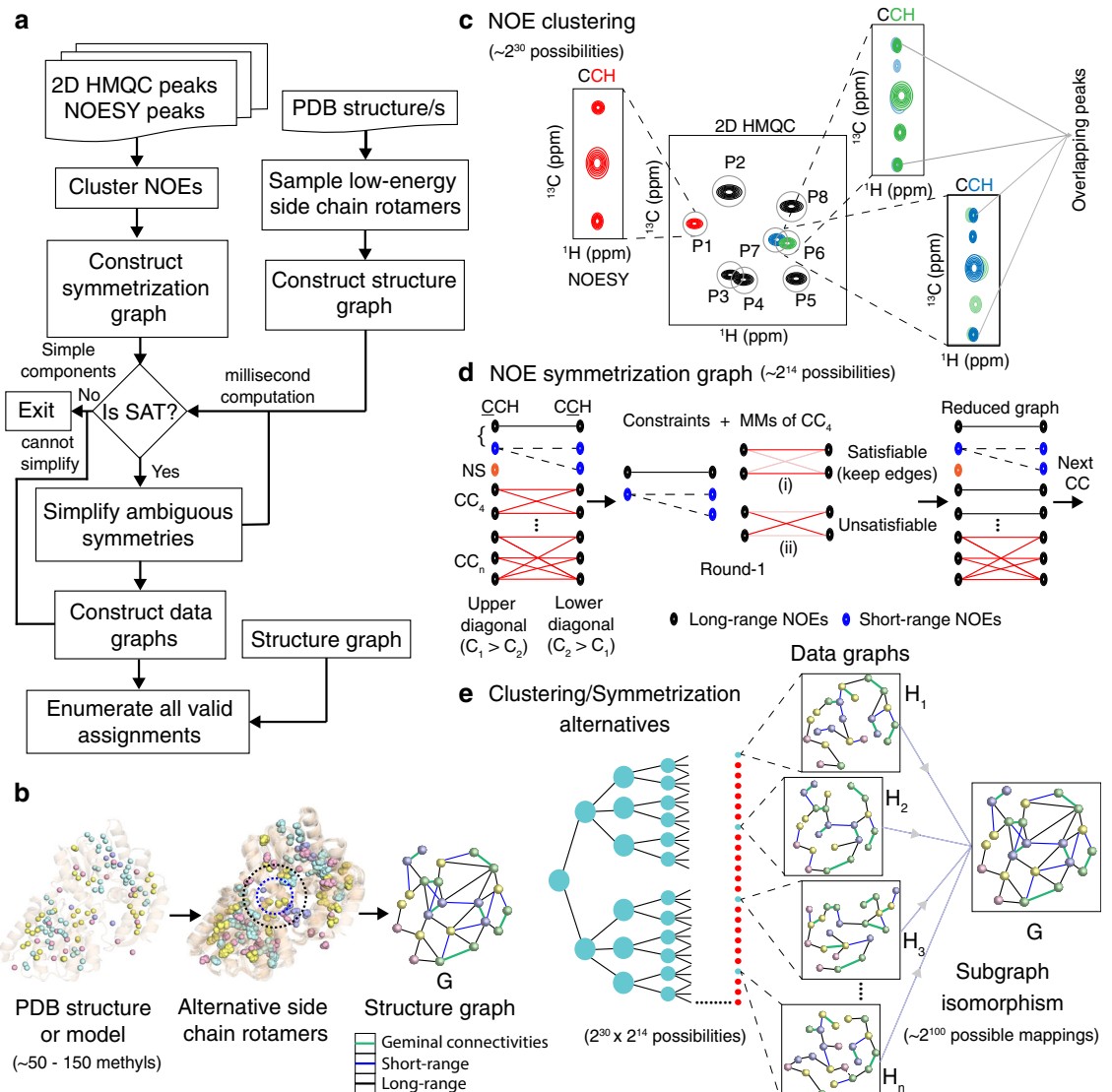

**Fig. 1 Exhaustive enumeration of methyl assignments from raw NOE peaks. a** Iterative workflow of the MAUS system. **b** Description of a structure graph, $G$ with (~50–150) methyls as nodes. The edges of $G$ correspond to all possible short-range (blue; up to 6–8 Å), long-range (black; from 6 up to 10 Å), and geminal methyl connectivities (green). **c** The 2D projections of 3D or 4D NOESY peaks are clustered (within tolerances; gray circles) to 2D HMQC reference peak positions (P1...P8). A NOESY peak can be clustered uniquely (red) or ambiguously (green and blue; overlapping peaks), leading to ~$2^{30}$ clustering combinations for typical 3D $C_M$-$C_M H_M$ NOESY data. **d** The observed chemical shift coordinates ($C_1,C_2,H_2$) of all NOE peaks are used to construct a symmetrization graph, $S$, with partitions representing upper ($C_1 > C_2$) and lower ($C_2 > C_1$) diagonal cross-peaks. $S$ has nodes represented by short-range (blue) and long-range (black) NOEs, and edges connecting potentially symmetric NOE peaks. $S$ has components of sizes 1 (no symmetry or NS), 2 and 3 (simple), and >3 (complex, CC), producing ~$2^{14}$ possibilities in the case of data recorded for maltose-binding protein, MBP. Simple components are used as constraints whereas complex components are reduced using an iterative process within MAUS: the maximum matchings (MMs) of each complex component are tested for satisfiability; satisfiable edges are retained in $S$. **e** A N-ary tree showing data graphs ($H_1,H_2,H_3,...,H_n$) generated from all clustering and symmetrization alternatives (~$2^{44}$ possibilities); a majority of these data graphs do not lead to satisfying assignments and are eliminated by MAUS (red circles). Each remaining $H$ can be mapped in ~$2^{100}$ possible ways onto $G$; all valid methyl resonance assignments are enumerated and presented to the user in a compact form.

obtained from the PDB and BMRB, respectively. To obtain a consistent set of experimental data for all targets, we prepared $^{13}$C/$^1$H (MA)ILV-methyl-labeled samples and acquired one 2D reference $^1$H-$^{13}$C heteronuclear multiple quantum coherence (HMQC) and two 3D $^{13}C_M$-$^{13}C_M{^1}H_M$ NOESY-HMQC spectra, recorded with short (50 ms) and long (300 ms) mixing times (Fig. 2, Supplementary Fig. 2, and Supplementary Table 2). Besides providing supplementary short-range (<6–8 Å) constraints, the short mixing time NOEs help reduce complex components of the symmetrization graph (Fig. 1d) and allow

determination of geminal connectivities for resonances corresponding to $^{13}C\gamma_1$/$^{13}C\gamma_2$ and $^{13}C\delta_1$/$^{13}C\delta_2$ methyls of Val and Leu residues, respectively. Finally, towards reducing spectral overlap in the 2D reference spectra of larger (>20 kDa) targets, we prepared proS-labeled samples, stereospecifically defining the resonances of Leu/Val methyls (Fig. 2). Using this information as input for MAUS, we find that among all possible clustering/symmetrization alternatives resulting from peak overlap ($2^{40}$-$2^{80}$), only a tractable number ($2^{10}$-$2^{20}$) can lead to valid assignment solutions (Supplementary Table 3) and are further explored in

**Table 1 MAUS methyl resonance assignment statistics.**

| Target | MW (kDa) | PDB ID | Origin/resolution (Å) | Number of methyls | Labeling scheme | Short-range NOE distance | % Unique assignments | % Options ≤ 3 and >1 |
|---|---|---|---|---|---|---|---|---|
| Hβ₂mᵃ | 11.9 | 1JNJ | NMR | 35 | AILV | 6.5 | 89 | 11 |
| HLA-A01 | 45.5 | 6AT9 | X-ray/2.9 | 87 | AILV | 7.2 | 64 | 30 |
| HLA-A02 | 44.8 | 1DUZ | X-ray/1.8 | 94 | AILV | 6.0 | 70 | 15 |
| MBP | 40.7 | 1DMB | X-ray/1.8 | 76 | MILVᵖʳᵒˢ | 6.0 | 75 | 21 |
| IL-2 | 15.4 | 1M47 | X-ray/1.9 | 61 | ILV | 6.5 | 74 | 23 |
| HNHᵇ | 15.7 | 6O56 | NMR/1.9 | 53 | ILV | 8.0 | 89 | 11 |
| REC2 | 15.6 | 4CMP | X-ray/2.6 | 69 | ILV | 6.5 | 59 | 12 |
| REC3 | 24.5 | 4ZT0 | X-ray/2.9 | 85 | ILV | 8.0 | 67 | 28 |

ᵃNon-stereospecific.
ᵇUsed symmetrization and clustering tolerances of 0.1 ppm for $^{13}$C.

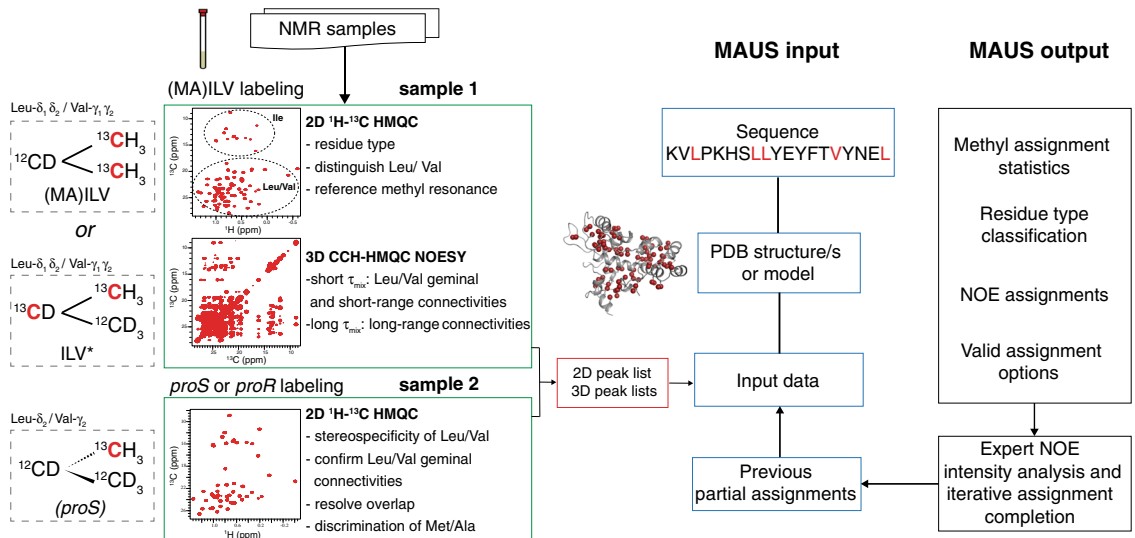

**Fig. 2 Sample preparation, data collection, and methyl assignment workflow.** Two separate protein samples are recommended for generating the standard inputs of MAUS. Sample **1** uses non-stereospecific methyl labeling with either (i) $^{13}$C/$^{1}$H labeling only at the methyls of Met, Ala, Leu, Val, and Ile residues, on a $^{12}$C/$^{2}$H background or (ii) $^{13}$C/$^{1}$H labeling at the methyls and $^{13}$C/$^{2}$H at the sidechains of methyl-bearing residues on a $^{12}$C/$^{2}$H background, creating linearized spin systems[43], which can be used to unambiguously distinguish Leu/Val methyl resonances[18] (Supplementary Figs. 4–6). When sample i is used, resonances of Leu/Val peaks can be partially distinguished using an automated chemical shift-based classifier within MAUS ("Methods"). Both samples can be used to define reference methyl chemical shifts using real-time (sample i) or constant-time (sample ii) 2D methyl-HMQC experiments. Long-range (up to 10 Å) NOEs are recorded using a 3D $C_M$-$C_MH_M$ SOFAST NOESY experiment (300 ms mixing time). A complementary NOESY experiment recorded with a short (typically 50 ms) mixing time is used to identify short-range NOEs, including between the geminal methyl resonances of Leu and Val residues. An additional protein sample **2** is prepared with stereospecific labeling (proS or proR) of Leu and Val methyls, which resolves spectral overlap in the 2D reference HMQC spectrum, and, together with sample **1**, allows unambiguous determination of geminal methyl pairs for Leu/Val. $^{13}$C nuclei are displayed in red. Steps of NMR data analysis are colored green, with the resulting data set illustrated as a red rectangle. The standard input files for MAUS (blue) are the primary sequence, the PDB coordinate file or model structure, the 2D $^{1}$H-$^{13}$C HMQC peak list, and two 3D $C_M$-$C_MH_M$ SOFAST NOESY peak lists (recorded with short and long mixing time). MAUS also accepts partial assignments that could be included in the user-annotated 2D $^{1}$H-$^{13}$C HMQC list, together with a specification of allowed residue types, stereospecificity, and geminal methyl connectivities (if present) for each 2D reference peak. The output includes methyl assignment statistics, residue type classification, NOE assignments, and final lists of assignment options. Although MAUS does not consider the NOE peak intensities, this information can be evaluated by the user toward further reducing assignment ambiguities in the output lists.

exhaustive subgraph isomorphism ($H$ into $G$) enumerations using the SAT algorithm (Fig. 1e). Given the sparsity of experimental NOE data sets and our definition of a valid solution, each subgraph isomorphism instance has up to $2^{100}$ valid solutions for a typical 100-methyl protein. However, the solution space is not uniformly distributed among methyl peaks; remarkably, results for all targets in our set show that the NOE network, residue type and stereospecificity constraints are sufficient to provide unambiguous assignments for a large fraction (64–89%) and low-ambiguity (two to three) options for the majority (11–30%) of remaining methyl resonances (Tables 1 and 2, and Supplementary Fig. 3).

**Robustness of MAUS to missing experimental inputs.** For optimal results, MAUS normally requires an NOE network with degree connectivity of 3.5 or greater (corresponding to an average of 1.75 experimentally observed NOE peaks per methyl). In order to unambiguously define the resonances of Leu and Val methyls and apply these as constraints to MAUS, we recommend the use of a homonuclear decoupling method[18] (Supplementary Fig. 4c, d). However, to account for cases where peak residue type information is not available experimentally, we also tested MAUS on all targets by either (i) defining resonances in the Leu/Val region as either Leu or Val, or (ii) engaging a chemical shift-based classifier, within

**Table 2 Performance comparison of different methyl assignment programs.**

| Target (labeling scheme) | No. of methyls | MAUS all defined | MAUS LV ambiguous | MAGIC all defined | MAGIC LV ambiguous | FLAMEnGO 2.4 | MAP-XSII | MethylFLYA |
|---|---|---|---|---|---|---|---|---|
| Hβ₂m (AILV) | 35 | 31/0 | 21/0 | 27/4 | 16/8 | 17/1 | 30/5 | ND[a] |
|  |  | 4/0 | 14/0 | 2/0 | 5/3 | 8/0 | 0/0 |  |
|  |  | 0/0 | 0/0 | 0/0 | 0/0 | 0/0 | 0/0 |  |
|  |  | 0 | 0 | 2 | 3 | 9 | 0 |  |
|  |  | 0% | 0% | 12% | 34% | 4% | 14% |  |
|  |  | 7 | 7 | 29 | 147 | 6 | 5 |  |
| HLA-A01 (AILV) | 87 | 56/0 | 48/0 | 58/4 | 50/11 | 56/3 | 79/8 | 64/0 |
|  |  | 26/0 | 30/0 | 17/1 | 11/0 | 1/0 | 0/0 | 0/0 |
|  |  | 5/0 | 9/0 | 3/0 | 2/4 | 0/0 | 0/0 | 0/0 |
|  |  | 0 | 0 | 4 | 9 | 27 | 0 | 0 |
|  |  | 0% | 0% | 6% | 15% | 5% | 9% | 0% |
|  |  | 15 | 18 | 296 | 422 | 23 | 12 | 704 |
| HLA-A02 (AILV) | 94 | 66/0 | 45/0 | 57/5 | ND[b] | 64/3 | 78/16 | 77/0 |
|  |  | 14/0 | 14/0 | 24/0 |  | 0/0 | 0/0 | 0/0 |
|  |  | 14/0 | 35/0 | 3/0 |  | 0/0 | 0/0 | 0/0 |
|  |  | 0 | 0 | 5 |  | 27 | 0 | 0 |
|  |  | 0% | 0% | 6% |  | 4% | 17% | 0% |
|  |  | 17 | 18 | 203 |  | 31 | 15 | 1024 |
| MBP (MILVproS) | 76 | 57/0 | 46/0 | 49/5 | ND[b] | ND[b] | 66/10 | 52/0 |
|  |  | 16/0 | 20/0 | 10/1 |  |  | 0/0 | 0/0 |
|  |  | 3/0 | 10/0 | 5/0 |  |  | 0/0 | 0/0 |
|  |  | 0 | 0 | 6 |  |  | 0 | 0 |
|  |  | 0% | 0% | 9% |  |  | 13% | 0% |
|  |  | 19 | 21 | 100 |  |  | 17 | 853 |
| IL-2 (ILV) | 61 | 45/0 | 6/0 | 23/23 | ND[b] | 21/4 | 36/25 | 23/2 |
|  |  | 14/0 | 32/0 | 5/1 |  | 0/0 | 0/0 | 0/0 |
|  |  | 2/0 | 23/0 | 2/0 |  | 0/0 | 0/0 | 0/0 |
|  |  | 0 | 0 | 7 |  | 36 | 0 | 0 |
|  |  | 0% | 0% | 46% |  | 16% | 41% | 9% |
|  |  | 5 | 9 | 451 |  | 30 | 13 | 2553 |
| HNH (ILV) | 53 | 47/0 | 27/0 | 39/0 | 38/0 | 29/2 | 38/15 | 44/0 |
|  |  | 6/0 | 18/0 | 12/2 | 3/2 | 4/0 | 0/0 | 0/0 |
|  |  | 0/0 | 8/0 | 0/0 | 6/4 | 0/0 | 0/0 | 0/0 |
|  |  | 0 | 0 | 0 | 0 | 18 | 0 | 0 |
|  |  | 0% | 0% | 4% | 5% | 6% | 28% | 0% |
|  |  | 4 | 4 | 50 | 79 | 15 | 12 | 814 |
| REC2 (ILV) | 69 | 41/0 | 37/0 | 21/3 | 26/7 | 23/4 | 28/35 | 7/2 |
|  |  | 8/0 | 12/0 | 20/4 | 14/5 | 0/0 | 0/0 | 0/0 |
|  |  | 20/0 | 20/0 | 6/3 | 3/1 | 0/0 | 0/0 | 0/0 |
|  |  | 0 | 0 | 12 | 13 | 42 | 6 | 0 |
|  |  | 0% | 0% | 15% | 23% | 15% | 56% | 29% |
|  |  | 5 | 5 | 87 | 218 | 35 | 7 | 814 |
| REC3 (ILV) | 85 | 57/0 | 28/0 | 58/0 | 54/2 | 47/3 | 61/24 |  |
|  |  | 24/0 | 28/0 | 19/0 | 19/3 | 8/0 | 0/0 | ND[a] |
|  |  | 4/0 | 29/0 | 7/0 | 6/0 | 0/0 | 0/0 |  |
|  |  | 0 | 0 | 1 | 1 | 27 | 0 |  |
|  |  | 0% | 0% | 0% | 6% | 5% | 39% |  |
|  |  | 14 | 14 | 58 | 131 | 51 | 16 |  |

*ND* not determined.
For results using a chemical shift-based classifier of Leu/Val resonances, see Supplementary Table 4.
First row: number of methyls with one option correct/wrong.
Second row: number of methyls with 2–3 options correct/wrong.
Third row: number of methyls with >3 options correct/wrong.
Fourth row: number of unassigned methyls.
Fifth row: error rate (in %).
Sixth row: run time in minutes.
[a]MethylFLYA returned an error.
[b]The simulations were allowed to run for at least 96 h but did not complete.

MAUS (see "Methods"). Removing Leu/Val residue type information has a significant impact on MAUS performance, resulting in an average of 27% decrease in assignment completeness across our benchmark set (Table 2, columns 3 and 4). Here, the use of the MAUS classifier to distinguish between confident Leu and Val methyl resonances led to an improved average of 18% less uniquely assigned peaks. Nonetheless, the accuracy of all assignments remained 100% and the ground truth solution was valid across all targets (Table 2 and Supplementary Table 4). For example, for MBP, MAUS assigned 71% of resonances uniquely and an additional 17% with two or three options while maintaining 100% accuracy, despite the absence of experimental residue type

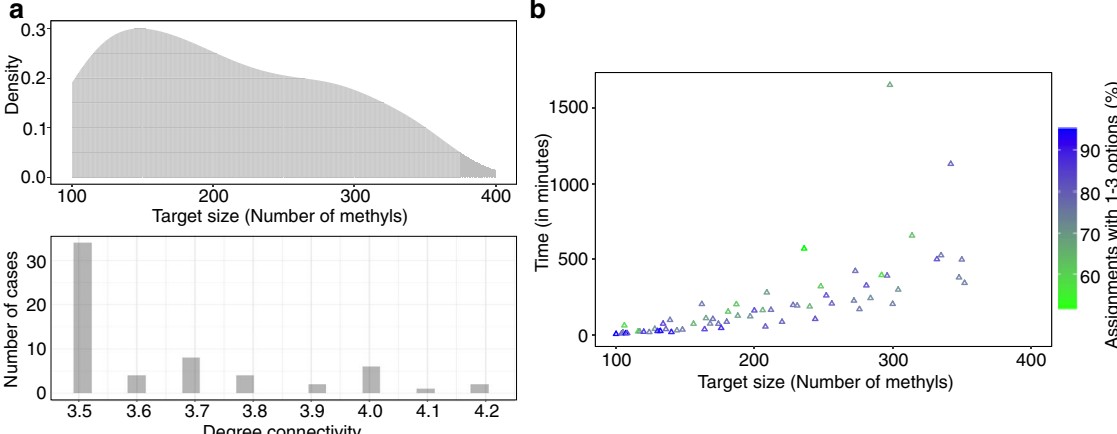

**Fig. 3 Performance of MAUS on simulated ILV data graphs from larger proteins. a** Top: target size distribution (in terms of number of methyl groups from Ile, Leu, and Val residues) of 63 proteins with high-resolution PDB structures of sizes up to 116 kDa (352 methyls; PDB ID: 5WTI) (top). For each target, a data graph $H$ was simulated by removing edges from the corresponding structure graph $G$ (defined as all methyl connectivities up to 10 Å present in the 3D structure) until a degree connectivity (defined as 2 × number of edges/number of nodes) in the range of 3.5–4.2 was reached, corresponding to 1.75–2.1 simulated NOEs/methyl. Bottom: bar plot showing the distribution of degree connectivities among all simulated data graphs. **b** Scatter plot showing run time (in minutes) taken by MAUS to perform exhaustive enumeration of the possible methyl assignment options for each $H$ into $G$ mapping case. Colors indicate % of methyls with up to three assignment options, according to the scale on the right.

information. In addition, if users cannot provide any annotations of the 2D reference peaks in the input, then MAUS inherently loses geminal connectivity constraints for all methyl resonances corresponding to Leu and Val residues, in addition to the residue type. It is therefore recommended, when using data sets where both geminal methyls are labeled, that MAUS is run using either missing geminal connectivity or Leu/Val residue type information, but not both.

**Performance of MAUS using simulated NOE networks from PDB structures**. Although results using our benchmark set of four targets with known assignments and recently recorded data highlight the utility of our method for proteins of sizes up to ~46 kDa, methyl-based NMR can be applied on much higher-molecular weight proteins. To test the size limit supported by MAUS, we expanded our simulation benchmark set to include 63 PDB structures with structure graphs containing more than 100 methyls from Ile, Leu and Val residues (corresponding to system sizes of up to 116 kDa) (Fig. 3). We chose to simulate assignment cases corresponding to the Isoleucine-Leucine-Valine (ILV), as opposed to the Alanine-Isoleucine-Leucine-Valine (AILV) labeling scheme, as a large fraction of the resonances corresponding to Ala methyls are missing from the spectra of larger proteins. For each target, a data graph was simulated by removing edges from the corresponding structure graph (defined as all methyl connectivities up to 10 Å present in the 3D structure) until a degree connectivity (defined as 2 × number of edges/number of nodes) of 3.5 or higher was reached. This value was chosen to represent experimental cases with 1.75–2.1 observed NOEs/methyl resonance, which are characteristic of real-life data sets. We find that, as long as this requirement is satisfied, MAUS can efficiently tackle the computational complexity of the graph-matching problem and provide meaningful assignments in a reasonable time (up to 4 h on a single central processing unit, CPU), even for larger targets. Specifically, our simulation results show that we can obtain a high coverage (60–80%) of methyl groups with 1–3 residue assignment options for targets up to 352 methyls. Moreover, MAUS can address proteins of up to 716 methyls/200 kDa (as exemplified by the Teneurin 2 partial extracellular

domain, PDB ID: 6FB3), albeit in a longer time (11 h on a single CPU), which is still feasible from a computational standpoint.

**De novo resonance assignments of Cas9 nuclease domains**. We further evaluated the performance of MAUS on deriving de novo, blind assignments for four representative targets; three domains of the Cas9 nuclease and the therapeutic cytokine interleukin-2 (IL-2) (15.4 kDa) (Fig. 4 and Supplementary Figs. 5–7). These comprise biomedically relevant proteins of mostly α-helical folds, leading to a high degree of spectral overlap, which challenges automated assignment methods. Cas9 is a 160 kDa RNA-guided endonuclease, which introduces DNA double-strand breaks upon site-specific recognition of a short nucleotide Protospacer Adjacent Motif, preceding the cleavage site[19]. The Cas9 enzyme comprises of a recognition lobe (REC) that forms an RNA:DNA hybrid through three subdomains (REC1–3), and a nuclease lobe with HNH and RuvC domains, which cleave the DNA strand that is complementary and non-complementary to the guide RNA, respectively (Fig. 4a, b). Towards establishing methyl assignments, we designed optimized constructs of individual Cas9 domains, HNH (15.7 kDa), REC2 (15.6 kDa), and REC3 (24.5 kDa), showing well-dispersed methyl NMR spectra (Fig. 4c). For these three domains, MAUS assigned 89%, 59%, and 67% of methyl resonances uniquely, and further provided assignment lists with two to three options for 11%, 12%, and 28%, respectively, while always maintaining 100% accuracy, relative to the ground truth assignments (Fig. 4d, Table 1, and "Methods"). We applied a selective-decoupling experiment to distinguish Leu/Val resonances[18] (Supplementary Figs. 6 and 7) and used conventional methods to obtain reference backbone and methyl assignments for validating the MAUS results (Supplementary Results). Concurrently with the development of the present work, the backbone and side-chain assignments for the HNH domain were obtained using a conventional backbone-based approach and released in the Biological Magnetic Resonance Data Bank (BMRB, entry 27949)[20,21]. The MAUS-derived methyl assignments were in full agreement with the published results, which further highlights the practical utility of our method in saving machine time and manual effort spent.

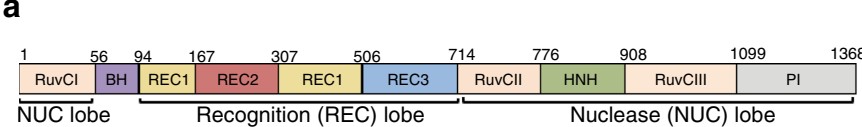

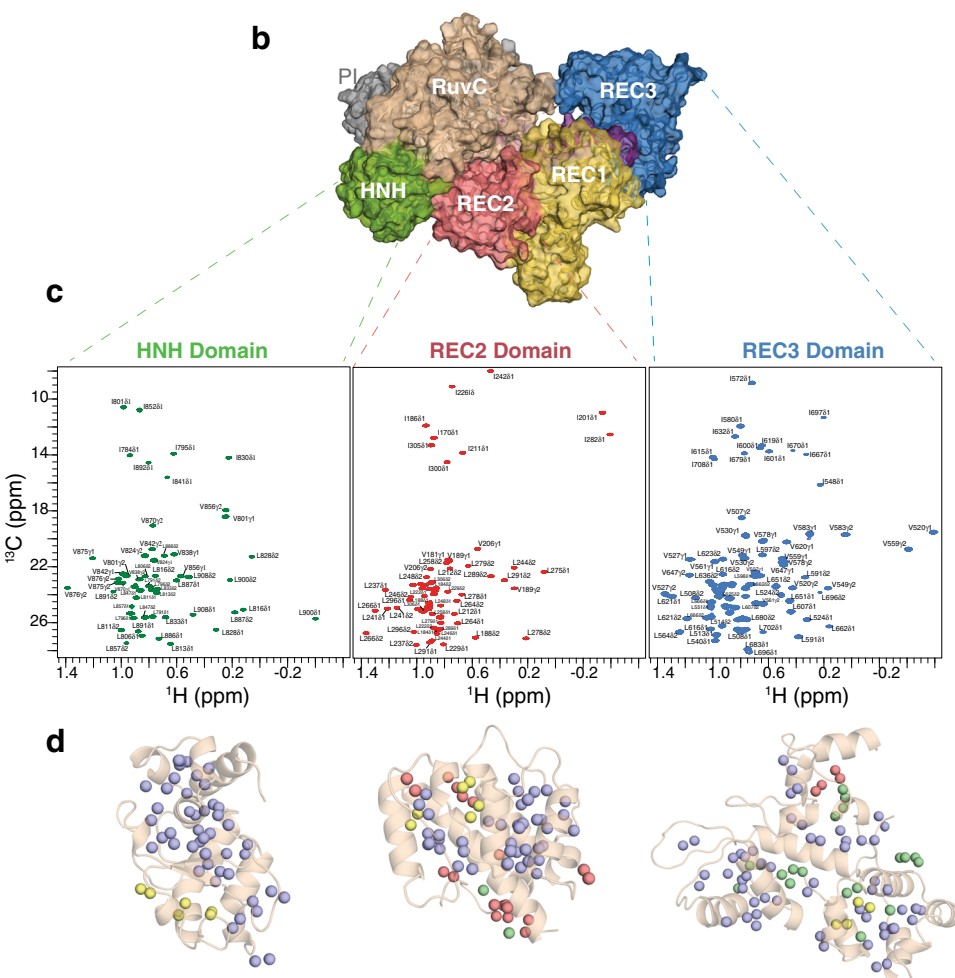

**Fig. 4 Streamlined resonance assignments of Cas9 domains using MAUS. a** Domain organization of *Spy*Cas9 composed of the recognition lobe (REC) and nuclease lobe (NUC). BH, bridge helix; PI, PAM-interacting. **b** Surface representation of SpyCas9 (PDB ID 4cmp) depicting the bilobed architecture. Protein domains are colored as in **a**. **c** Single domains used in the divide-and-conquer approach. $^1H$-$^{13}C$ HMQC spectra of selectively labeled HNH (green), REC2 (red), and REC3 (blue) domains at Ile $\delta_1$-$^{13}CH_3$; Leu, Val-$^{13}CH_3$/$^{13}CH_3$ positions, acquired at 800 MHz, 25 °C. The spectra of the individual domains of Cas9 indicate that they retain their proper fold when in isolation. **d** Number of valid assignment options for each residue identified by MAUS for HNH, REC2, and REC3, respectively. The colored spheres represent final valid resonance assignment options: violet (1 option), green (2 options), yellow (3 options), and red (>3 options).

For IL-2, MAUS identified unique assignments for 74% and provided a short list (two to three) of confident options for an additional 23% of all methyl resonances (Table 1 and Supplementary Fig. 3). With the MAUS lists in hand, we manually considered the NOE peak intensities to reach complete (>95%) assignments for all targets in a matter of hours. Our results show that, together with the quality of the NMR sample, the protein fold and methyl chemical shift dispersion is equally important for obtaining complete assignments. Notably, the amide $^{15}N$-$^1H$ TROSY spectrum of the Cas9 REC3 domain exhibits a significant fraction of missing backbone amide resonances (~55%, residues 660–712), likely due to conformational exchange-induced line broadening. Thus, methyl-based NMR supported by MAUS, is a

practically useful tool to study not only larger systems with intractable backbone amide spectra, but also enabling routine applications for the study of medium-sized targets.

**Comparison with previous methods.** When compared with other methyl assignment tools (MAGIC/FLAMEnGO2.4/MAP-XSII) using the same inputs for all targets with known assignments in our benchmark set, MAUS produces 100% accurate results, while the previous methods show error rates of up to (34/16/56%), and significantly lower assignment completeness rates (Table 2 and Supplementary Table 5). Consistently with the results obtained for our benchmark set, MAUS maintains an

improved performance on the 4 blind targets in our set (IL-2, REC2, HNH, and REC3), in terms of both unambiguous assignments and error rate (Table 2).

A recent commercially available method, MethylFLYA[12], outperforms the existing publicly available methyl assignment tools. We carried out a detailed comparison of MAUS (also readily available to users via an online interface) with Methyl-FLYA on the eight targets used in our study, by providing the identical inputs to both methods. Upon comparison of the number of resonances assigned uniquely by MAUS and with high confidence by MethylFLYA, we find that MAUS outperforms MethyFLYA for 6/8 targets in our set (Hβ$_2$m, MBP, IL-2, HNH, REC2, and REC3). Specifically, (i) for IL-2 and REC2, Methyl-FLYA reports a significantly lower number of confident assignments (38% and 10%) relative to MAUS (74% and 60%), while also producing two incorrect assignments for each target, (ii) for HNH and MBP, it produces confident assignments at lower completeness levels, and (iii) for Hβ$_2$m and REC3, MethylFLYA crashes (segmentation fault), producing no results. Although MethylFLYA assigns a higher fraction of methyl resonances for the two larger targets in our set (HLA-A01 and HLA-A02), it requires a prohibitively longer computation time on a single CPU and, in general, it is less efficient than MAUS by two to three orders of magnitude (Table 2). Finally, although MethylFLYA provides low-confidence options for some of the unassigned methyl resonances, MAUS provides a list of valid options for each 2D reference peak, which contains the correct assignment, owning to the fact that it applies a satisfiability-based enumeration process that is both exact, and exhaustive.

**Consistent methyl assignments guided by sequence-based models.** The structures of targets which lack a representative template in the PDB can be modeled using homology-based methods[22] or, in cases with no identifiable sequence homologs in the PDB, sequence co-evolution approaches[23,24]. To examine the latter, more challenging case, we used trRosetta (transform-restrained Rosetta)[25], which has demonstrated high accuracy for complex Critical Assessment of Structure Prediction[26] targets. We derived a set of models for a representative case in our benchmark set, MBP using the trRosetta web-server, and used this together with our experimental NOE data recorded on a MILV$^{proS}$ sample (Supplementary Table 2) to run MAUS. The models showed a 3.3 Å average all-atom root-mean-square deviation (RMSD) from the X-ray structure, nonetheless, MAUS still provided unique and low-ambiguity (two to three options) assignments for 68% of resonances (relative to 96% using the X-ray structure), while maintaining 100% accuracy (Fig. 5a).

As structure prediction methods may also lead to globally incorrect models, we examined whether MAUS can distinguish between two closely related structures of the same protein sequence directly from unprocessed 3D NOE peaks (picked directly from 3D NOESY data using a signal-to-noise (S/N) ratio ≥ 5). To test this, we used a designed homotrimeric protein (XAA) consisting of a multi-layer, three-helix bundle with modular termini that may adopt two divergent structures corresponding to an open (observed by X-ray) and a closed state, which corresponds to the solution structure determined using manual NOE assignments by our group[27]. Starting from the unassigned NOE peaks, we ran MAUS using both models as inputs including both intra- and inter-chain connectivities in the structure graph $G$, and found that only the closed state led to assignments which satisfy the global NOE network within our default 10 Å upper limit. MAUS finds that the data graph is not isomorphic to the structure graph derived from the open state,

due to a subset of NOE constraints that are violated by more than 25 Å (Fig. 5b). This result justifies our choice of an exact subgraph isomorphism approach, since any attempt to arbitrarily remove NOEs (i.e., edges of $H$) not explained by the input structure (i.e., edges of $G$) could lead to a cascade effect, resulting to global changes in the space of solutions.

In summary, we demonstrate that a satisfiability-based approach can deliver reliable assignments for a range of targets amenable to solution-state NMR, using unprocessed NOE peak data (3D or 4D peak lists) and an existing structural model with the correct overall fold. Our results using NOESY spectra recorded for proteins with known assignments as well as for several blind targets show that, unlike previous methods, MAUS provides 100% accurate solutions for a large fraction (60–80%) of methyl groups, thereby reducing manual effort, costs, and errors introduced due to manual pre-processing and validation of the data. We further demonstrate that, in the absence of high-resolution structures or structural homologs in the PDB, sequence co-evolution-based models can be used by MAUS, without compromising the correctness of produced assignments. Alternatively, for larger, multi-domain proteins with complex methyl spectra, a user may apply a divide-and-conquer strategy, supported by MAUS. Using our online web-server, users can now assign the methyl spectra of large, multi-domain proteins in a matter of a few days, also considering the time it takes to record the NMR data. Our method opens new possibilities for studying challenging, complex molecular machines, as illustrated here for the Cas9 nuclease.

## Methods
**Protein expression and purification of benchmark targets.** DNA corresponding to the luminal domains of Hβ$_2$m, HLA-A*01:01, and HLA-A*02:01 expressed in *Escherichia coli* BL21 (DE3) cells, refolded and purified as described[27,28]. Briefly, Hβ$_2$m, HLA-A*01:01, and HLA-A*02:01 induced with 1 mM isopropyl β-D-1-thiogalactopyranoside (IPTG) at an OD600 of 0.7 at 37 °C for 4 h, isolated from inclusion bodies and refolded in vitro. Inclusion bodies were isolated from *E. coli* cell pellets by sonication, following by a wash with 100 mM Tris pH 8, 2 mM EDTA, 0.1% (v/v) deoxycholate, and solubilization in 5.5 M guanidine hydrochloride under reducing conditions. For in vitro refolding of Hβ$_2$m, 100 mg protein was slowly diluted dropwise over 24 h into refolding buffer (0.4 M arginine-HCl, 2 mM EDTA, 4.9 mM reduced L-glutathione, 0.57 mM oxidized L-glutathione, 100 mM Tris pH 8.0) at 4 °C while stirring. For in vitro refolding of HLA-A*01:01 and HLA-A*02:01, 10 mg of peptide (ILDTAGKEEY for HLA-A*01:01 and LLFGYPVYV for HLA-A*02:01), and 100 mg heavy chain and 100 mg Hβ$_2$m were slowly diluted dropwise over 24 h into refolding buffer (0.4 M arginine-HCl, 2 mM EDTA, 4.9 mM reduced L-glutathione, 0.57 mM oxidized L-glutathione, 100 mM Tris pH 8.0) at 4 °C while stirring. All refolding proceeded for 4 days at 4 °C without stirring. Purification of Hβ$_2$m, HLA-A*01:01, and HLA-A*02:01 was performed by size-exclusion chromatography (SEC) with a HiLoad 16/600 Superdex 75 pg column at 1 mL/min with running buffer (150 mM NaCl, 25 mM Tris pH 8).

XAA was expressed in *E. coli* BL21 Star (DE3) cells and purified as described[26]. Briefly, XAA was induced with 0.1 mM IPTG at an OD600 of 0.6 and expressed at 18 °C for 17 h. Cells were resuspended in lysis buffer (25 mM Tris pH 8, 300 mM NaCl, 20 mM imidazole) and lysed by sonication. Supernatant was applied to nickel-nitrilotriacetic acid (Ni-NTA) resin pre-equilibrated with lysis buffer. The column was rinsed with ten column volumes of wash buffer (25 mM Tris pH 8, 300 mM NaCl, 20 mM imidazole). XAA was eluted using two column volumes elution buffer (25 mM Tris pH 8, 300 mM NaCl, 250 mM imidazole). The hexahistidine tag was removed via thrombin cleavage by incubating with 1 : 5000 thrombin (EMD, Millipore) for 15 h at 25 °C. Cleaved XAA was purified by SEC using a Superdex 75 10/300 GL column (GE Healthcare) at 1 mL/min with running buffer (150 mM NaCl, 25 mM Tris pH 8).

The MBP-cyclodextrin complex was prepared as described previously[29]. Briefly, MBP was expressed in *E. coli* strain BL21(DE3) cells, induced with 1 mM IPTG at an OD600 of 0.7 and expressed at 37 °C for 5 h. Purification of MBP-cyclodextrin was achieved as follows. First, MBP was bound to an amylose affinity column in the presence of binding buffer (150 mM NaCl, 25 mM Tris pH 8, 1 mM EDTA) and then eluted using eluted buffer (150 mM NaCl, 25 mM Tris pH 8, 1 mM EDTA, 10 mM maltose). Second, MBP was further purified by SEC with a HiLoad 16/600 Superdex 75 pg column at 1 mL/min with running buffer (150 mM NaCl, 25 mM Tris pH 8). MBP was then partially unfolded for 3 h at 25 °C in 150 mM NaCl, 25 mM Tris pH 8, 2.5 M guanidinium hydrochloride, and refolded by dilution into GuHCl-free buffer containing 2 mM β-cyclodextrin. Following purification all

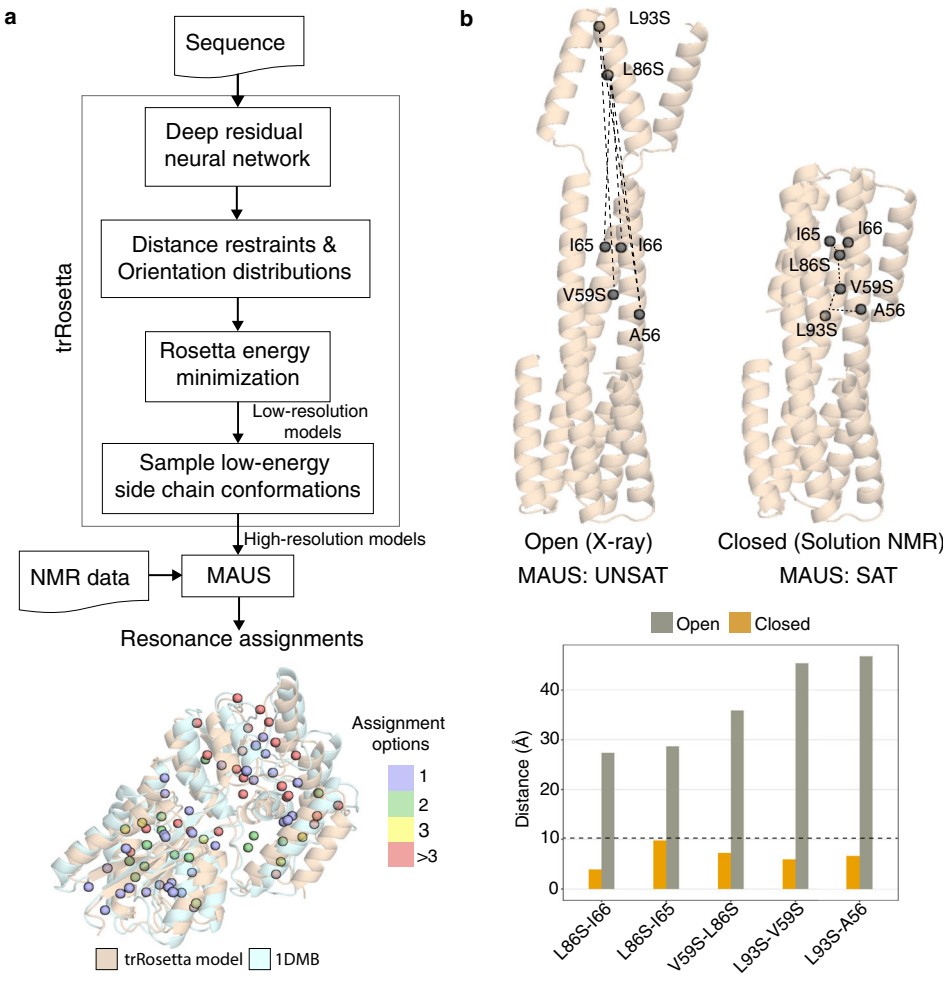

**Fig. 5 Robustness of MAUS assignments from different structural models. a** Top: flowchart showing the combined de novo structure modeling and methyl resonance assignment strategy using trRosetta (transform-restrained Rosetta) and MAUS, respectively. (bottom) Overlay of trRosetta model (wheat) and X-ray structure (PDB ID: 1DMB; pale cyan) of maltose-binding protein. Number of resonance assignment options identified by MAUS are indicated by colored spheres (violet: 1, green: 2, yellow: 3 and red: >3). **b** Top: structures of a designed homotrimeric protein[27], in the open (solved by X-ray crystallography) and closed (solved by solution NMR) states showing inter-methyl distances corresponding to experimental NOEs identified in a 3D $C_M$-$C_M H_M$ NOESY spectrum, assigned manually[27]. Gray spheres show methyl residues participating in the NOE network. The dashed lines represent pairwise distances in the two models, also shown as bar plots (bottom) for the open (gray) and closed (golden) states. The dashed line in the bar plot highlights the 10 Å NOE upper distance limit used by MAUS. Using the unassigned NOE peaks, MAUS delivers a set of satisfying assignments when provided with a model of the closed, but not of the open structure as input.

proteins were exhaustively buffer exchanged in their respective buffers (see section "Stereospecific isotopic labeling").

***Streptococcus pyogenes* Cas9 domain constructs**. The coding sequences for Cas9 HNH (residues 776–908) and REC2 domains (residues 167–307) were synthesized, codon optimized for expression in *E. coli*, and subcloned into a His$_{10}$-MBP expression vector (Genscript) (Supplementary Table 6). pSHS325 bacterial expression plasmid for SpCas9 REC3 domain was a gift from Jennifer Doudna & Keith Joung (Addgene plasmid #101205)[30]. Each protein was expressed in *E. coli* BL21 (DE3) containing chaperone plasmid pG-KJE8 (TAKARA, 3340) to enhance protein folding[31] and purified as described below[32]. Briefly, when cells reached an OD600 of ~0.6, IPTG was added to a final concentration of 0.5 mM to induce protein expression. Cells were then grown for an additional 18 h at 23 °C. Collected cells were resuspended in lysis buffer (50 mM Tris pH 7.5, 500 mM NaCl, 5% (v/v) glycerol, and 1 mM Tris(2-carboxy-ethyl) phosphine (TCEP) containing an EDTA-free protease inhibitor tablet (Roche). The cell suspension was sonicated on ice and clarified by centrifugation at 27,000 × *g* for 15 min. The soluble lysate fraction was bound in batch to Ni-NTA agarose (Qiagen). The resin was washed extensively with 20 mM Tris pH 7.5, 500 mM NaCl, 1 mM TCEP, 10 mM imidazole, and 5% (vol/vol) glycerol, and the bound protein was eluted in 20 mM Tris pH 7.5, 500 mM NaCl, 1 mM TCEP, 300 mM imidazole, and 5% (vol/vol) glycerol. The His$_{10}$-MBP affinity tag was removed with His$_{10}$-tagged TEV protease during overnight dialysis against 20 mM Tris pH 7.5, 500 mM NaCl, 1 mM TCEP, and 5% (vol/vol) glycerol. The protein was then flowed over Ni-NTA agarose to remove

TEV (Tobacco Etch Virus) protease and the cleaved affinity tag, and further purified by SEC on a Superdex 200 16/60 column (GE Healthcare) in 20 mM Tris pH 7.5, 200 mM KCl, 1 mM TCEP, and 5% (vol/vol) glycerol.

**Human IL-2 expression, refolding, and purification**. Codon optimized DNA encoding the human IL-2 (amino acids 1–133) with a site-specific mutation (C140S) (Supplementary Table 6) was expressed in BL21 (DE3) *E. coli* cells as inclusion bodies. Protein expression was achieved by induction with 1 mM IPTG at an OD600 of 0.6 followed by cell growth at 37 °C for 5 h at 200 r.p.m. For in vitro refolding, ~30 mg of inclusion bodies was dropped diluted into 200 mL of refolding buffer (1.1 M guanidine, 6.5 mM cysteamine, 0.65 mM cystamine, 110 mM Tris pH 8.0) at 4 °C while stirring. Refolding proceeded overnight at 4 °C without stirring. The solution was dialyzed into a buffer of 20 mM MES pH 6.0, 25 mM sodium chloride. Purification of refolded IL-2 was performed by cation exchange chromatography with a CAPTO-SP column using a 25 mM to 1 M NaCl gradient in a buffer with 25 mM MES pH 6.0 followed by SEC with a Superdex 75 column (GE) at 0.5 mL/min with running buffer (50 mM NaCl, 20 mM sodium phosphate pH 6.0). Protein concentrations were determined using A$_{280}$ measurements on a NanoDrop with extinction coefficients estimated with the ExPASy ProtParam tool.

**Preparation of NMR samples, backbone, and methyl assignments**. All proteins were overexpressed in M9 medium in $^2H_2O$ containing 2 g l$^{-1}$ $^2H^{13}C$ glucose (Sigma #552151) and 1 g l$^{-1}$ $^{15}NH_4Cl$. Selective methyl labeling, referred to as ILV*, was achieved by the addition of appropriate precursors (ISOTEC Stable

Isotope Products (Sigma-Aldrich) as detailed previously[28,33]. ILV-methyl (Ile $^{13}C\delta_1$; Leu $^{13}C\delta_1/^{13}C\delta_2$; Val $^{13}C\gamma_1/^{13}C\gamma_2$ U-[$^{15}$N, $^2$H]-labeled proteins were prepared in M9 medium in $^2$H$_2$O, supplemented with 2 g l$^{-1}$ $^2$H$^{12}$C glucose (Sigma #552003) and 1 g l$^{-1}$ $^{15}$NH$_4$Cl. De novo stereospecific methyl assignments were achieved by utilizing three independently prepared isotopically labeled samples: AILV or ILV (Ala $^{13}C\beta$; Ile $^{13}C\delta_1$; Leu $^{13}C\delta_1/^{13}C\delta_2$; Val $^{13}C\gamma_1/^{13}C\gamma_2$ in an otherwise U-[$^{15}$N, $^{12}$C, $^2$H] background), ILV$^{proS}$ (Ile $^{13}C\delta_1$; Leu $^{13}C\delta_2$; Val $^{13}C\gamma_2$ in an otherwise U-[$^{15}$N, $^{12}$C, $^2$H] background), and ILV* (Ile $^{13}C\delta_1$ only; Leu $^{13}C\delta_1/^{13}C\delta_2$; Val $^{13}C\gamma_1/^{13}C\gamma_2$ in an otherwise U-[$^{15}$N, $^{13}$C, $^2$H] background). We employed a multipronged approach where backbone assignments are used to aid the assignment of side-chain methyl groups. Specifically, we first obtained backbone assignments using TROSY-based 3D HNCA, HN(CA)CB, and HNCO experiments recorded with the ILV*-labeled samples. Acquisition times of 30 ms ($^{15}$N), 14 ms ($^{13}$CO), and 10/5 ms ($^{13}$C$\alpha$/$^{13}$C$\beta$) in the indirect dimension were used. Backbone amide assignments were confirmed using amide-to-amide NOEs obtained from 3D H$_M$-NH$_N$ and 3D N-NH$_N$ SOFAST NOESY-HMQC experiments[7]. Final backbone assignments were further validated using TALOS-N. Next, Ala, Ile, Leu, and Val methyl assignments were achieved using methyl-to-methyl NOEs observed in 3D H$_M$-C$_M$H$_M$ and 3D C$_M$-C$_M$H$_M$ SOFAST NOESY-HMQC in addition to methyl-to-amide NOEs observed in 3D H$_N$-C$_M$H$_M$ and 3D C$_M$-NH$_N$ SOFAST NOESY-HMQC experiments[7] recorded on the AILV- or ILV-labeled samples. For the 3D H$_N$-C$_M$H$_M$ SOFAST NOESY, the acquisition parameters were 64, 48, and 1280 complex points in the $^1$H$_N$, $^{13}$C$_M$, and $^1$H$_M$ dimensions with corresponding acquisition times of 14.5, 10.8, and 79 ms with eight scans/FID. For the 3D N-C$_M$H$_M$ SOFAST NOESY, the acquisition parameters were 48, 40, and 1280 complex points in the $^{15}$N, $^{13}$C$_M$, and $^1$H$_M$ dimensions with corresponding acquisition times of 26, 9, and 79 ms with eight scans/FID (free induction decay), respectively.

Leu/Val geminal pairs were determined by comparing NOE strips in 3D C$_M$-C$_M$H$_M$ SOFAST NOESY experiments recorded using short (50 ms) and long (300 ms) mixing times. The acquisition parameters for the 3D H$_M$-C$_M$H$_M$ SOFAST NOESY and 3D C$_M$-C$_M$H$_M$ SOFAST NOESY are provided in Supplementary Table 2. Ile $\delta_1$ methyl types were identified by their characteristic upfield chemical shifts. Leu and Val methyl types were identified using phase-sensitive 2D $^1$H-$^{13}$C HMQC experiments recorded on the ILV*-labeled sample[18]. Alanine methyl types were identified by comparing 2D $^1$H-$^{13}$C HMQC spectra of AILV and ILV$^{proS}$ samples, and by comparing Ala C$_\beta$ chemical shifts from HN(CA)CB experiments. Finally, stereospecific Leu $\delta_2$ and Val $\gamma_2$ methyl assignments were obtained by comparison of 2D $^1$H-$^{13}$C HMQC spectra of AILV- and ILV$^{proS}$-labeled samples. All the data were recorded at a $^1$H field of 750 or 800 MHz at 25 °C. The 3D SOFAST NOESY experiments were recorded at 800 MHz, 25 °C using a recycle delay of 0.2 s, and NOE mixing time of 50 ms and 300 ms. Typical data acquisition times were 1 h for the 2D HMQC, 8–10 h for the 50 ms, and 12–14 h for the 200 ms 3D SOFAST NOESY-HMQC experiments. All spectra were acquired using Topspin 3 acquisition software from Bruker. The data were processed in NMRPipe[34] and analyzed in NMRFAM-SPARKY[35] and CcpNMR[36] programs.

**Stereospecific isotopic labeling.** A specifically methyl-labeled acetolactate precursor (2-[$^{13}$CH$_3$], 4-[$^2$H$_3$] acetolactate) was obtained through deprotection and exchange of the protons of the methyl group in position four of ethyl 2-hydroxy-2-($^{13}$C)methyl-3-oxobutanoate (FB reagents) achieved in D$_2$O and pH 13[37]. Typically, 300 mg of ethyl 2-hydroxy-2-($^{13}$C)methyl-3-oxobutanoate was added to 24 mL of a 0.1 M NaOD/D$_2$O solution. After 30 min, the solution was adjusted to neutral pH with DCl and 2 mL of 1 M Tris pH 8.0 in D$_2$O was added. For the production of highly deuterated [U-$^2$H], I-[$^{13}$CH$_3$]$\delta_1$, L-[$^{13}$CH$_3$]proS, V-[$^{13}$CH$_3$]proS samples, 300 mg/L of 2-[$^{13}$CH$_3$], 4-[$^2$H$_3$] acetolactate, prepared as described above, was added 1 h prior to induction (OD600 $\approx$ 0.55). Forty minutes later (i.e., 20 min prior to induction), 3,3-[$^2$H$_2$],4-[$^{13}$C]-2-ketobutyrate (SIGMA #589276) was added to a final concentration of 60 mg/L. Each protein was induced as described above.

For each labeled protein sample, the concentration and buffer composition was as follows:

- 0.5 mM HLA-A01 in 20 mM sodium phosphate (pH 7.2), 50 mM NaCl, 5% D$_2$O.
- 0.5 mM HLA-A02 in 20 mM sodium phosphate (pH 7.2), 50 mM NaCl, 5% D$_2$O.
- 1.3 mM Hβ$_2$m in 20 mM sodium phosphate (pH 7.2), 50 mM NaCl, 5% D$_2$O.
- 0.5 mM XAA domain in 20 mM sodium phosphate (pH 6.2), 100 mM NaCl, 5% D$_2$O.
- 0.8 mM MBP domain in 20 mM sodium phosphate (pH 7.2), 50 mM NaCl, 5% D$_2$O.
- 2.0 mM HNH domain in 20 mM HEPES (pH 7.5), 80 mM KCl, 5% D$_2$O.
- 0.5 mM REC2 domain in 20 mM sodium phosphate (pH 7.2), 50 mM NaCl, 5% D$_2$O.
- 0.7 mM REC3 domain in 20 mM HEPES (pH 7.5), 200 mM KCl, 5% deuterated glycerol-d8, 1 mM TCEP, 0.01% NaN$_3$, 5% D$_2$O.
- 0.4 mM IL-2 in 20 mM sodium phosphate (pH 6.0), 50 mM NaCl, 5% D$_2$O.

**Generating a structure graph G in MAUS system.** The input PDB structure is utilized by MAUS to construct an undirected graph G (Fig. 1b). First, for all pairs of methyls in the input structure, we compute the following function of the average

distance between all pairs of protons from the two methyl groups (Eq. (1)):

$$\Delta_s(p,q) = \left( \frac{1}{9} \sum_{i=1}^{3} \sum_{j=1}^{3} d\left(p_i, q_j\right)^{-6} \right)^{-\frac{1}{6}} \quad (1)$$

Where $p$ and $q$ are methyl groups and $p_i$ and $q_j$ represent protons of each methyl group in an input structure $s$, and $d$ denotes the Euclidean distance. When calculating an effective distance between the nine pairs of methyl protons, the average value of the distances (i.e., multiplicity correction) used here results in a minor increase in upper distance bounds relative to more commonly used $r^{-6}$ summation. For instance, if we have three distance measures 4 Å, 5.5 Å, and 7 Å, the regular $r^{-6}$ summation results in a value of 3.9 Å, whereas the average value is 4.7 Å, which remains within the range of the observed interproton distances. Due to the fact that MAUS explicitly considers different side-chain rotamers in addition to applying relatively large upper distance thresholds, the exact choice of distance averaging does not influence our results significantly (as opposed to a more precise estimate of upper bounds that is required for NMR structure determination applications).

To account for alternative side-chain rotamers, the input structure is subjected to $n$ (quick: 10 or thorough: 100) independent relaxations using the FastRelax[38] protocol in Rosetta. Therefore, the element of the adjacency matrix $\Delta(p, q)$ is defined by taking the minimum of the distance functions observed among all sampled conformations (Eq. (2)):

$$\Delta(p,q) = \min_{s \in S} \Delta_s(p,q) \quad (2)$$

We call an edge of the structure graph, G (i) geminal, if it corresponds to a connectivity between the $\gamma_1/\gamma_2$ and $\delta_1/\delta_2$ methyls of Leu and Val residues, (ii) long-range if 6 Å < $\Delta(p, q) \leq$ 10 Å, and (iii) short-range if $\Delta(p, q) \leq$ 6 Å.

**Deriving all possible data graphs, H, from the input NOE data.** The two 3D NOESY spectra (recorded with short and long mixing times) are picked at a typical S/N level of 5 or higher and provided as an input list of (C1, C2, H2) coordinates. In addition, a high-resolution 2D reference methyl-HMQC spectrum is picked to identify all reference (C, H) methyl resonances. For larger (>20 kDa) protein targets, simultaneous consideration of the 2D spectrum from a separate, stereospecifically labeled proS sample helps resolve overlap, to identify the exact $^{13}$C and $^1$H chemical shifts corresponding to each observable methyl in the NMR sample. The 2D reference and two 3D NOESY peak lists are provided as inputs to MAUS.

We model all alternative NOE connectivities resulting from overlap in the input 2D and 3D spectra, to allow an explicit consideration of all possible data graphs, H, which are consistent with the input data. First, a 3D NOESY cross-peak can be projected to one or several 2D reference peaks (Fig. 1c). In particular, a 3D maximum (C1, C2, H2) can be projected to a 2D maximum (C, H) if and only if the following holds true (Eqs. (3) and (4)):

$$|C2 - C| < C_{tol}^{cluster} \quad (3)$$

$$|H2 - H| < H_{tol}^{cluster} \quad (4)$$

The tolerance values are set, by default, to 0.15 p.p.m. for $^{13}$C and 0.02 p.p.m. for $^1$H chemical shifts. NOE peaks with no identifiable projection, as well as diagonal NOEs are eliminated. This process yields clusters of tentative NOE connectivities for each peak in the 2D reference set.

Second, each valid (true positive) connectivity present in the input data must arise from one upper-diagonal NOE cross-peak with one corresponding, symmetry-related lower-diagonal counterpart. However, identifying unique symmetry relations between pairs of 3D NOE cross-peaks can be challenging due to spectral overlap. Specifically, two 3D NOE peaks (C1, C2, H2) and (C1', C2', H2') are potentially symmetric (Eqs. (5) and (6)) if and only if

$$|C2 - C1'| < C_{tol}^{sym} \quad (5)$$

$$|C1 - C2'| < C_{tol}^{sym} \quad (6)$$

To explore all possible symmetry relations present in the input data, we construct a bipartite symmetrization graph connecting upper-diagonal to lower-diagonal NOE cross-peaks (Fig. 1d). An edge appears between all pairs of peaks that are potentially symmetric. For typical NOE data sets, the bipartite symmetrization graph consists of several connected components (Fig. 1d). Isolated peaks are eliminated. Components of sizes 2 and 3, termed simple, provide edges of the data graph, H. Larger, complex components are iteratively reduced to smaller ones using explicit satisfiability, eliminating edges that do not yield satisfying assignments in exhaustive enumerations, as outlined in detail below (Fig. 1d).

**Reducing subgraph isomorphism to satisfiability.** A boolean function $f$ on a set of boolean variables $x = \{x_1, x_2, x_3 ..., x_n\}$ is said to be satisfiable if there exists an assignment $x \in \{0, 1\}^n$ such that $f(x)$ evaluates to 1. The question of whether a function $f$ is satisfiable is known as the satisfiability (SAT) problem in computer science. A boolean function $f$ is in conjunctive normal form (CNF) if it is a conjunction of one or more clauses, where a clause is a disjunction of literals; in other words, it is an AND of ORs. Although satisfiability is NP-complete, i.e., theory suggests that no polynomial-time algorithm exists, in practice, efficient

satisfiability solvers have been developed, solving formulas with millions of variables and clauses from milliseconds to a few seconds[39].

The resonance assignment problem is a subgraph isomorphism problem, which is also NP-complete[13]. MAUS reduces resonance assignment to a boolean satisfiability problem, exploiting the power of satisfiability solvers to explore the space of all possible assignment solutions exhaustively. Specifically, given an instance of a resonance assignment problem, we construct a boolean formula introducing the following variables:

1. $X(i, j)$: a boolean variable declaring whether 2D peak $i$ is assigned to methyl $j$ in the input structure.
2. $Y(k, l)$: a boolean variable declaring whether NOE peak $k$ is clustered to 2D peak $l$.
3. $Z(k, k')$: a boolean variable declaring whether NOE peaks $k$ and $k'$ are truly symmetric.

Next, MAUS combines these variables into a CNF formula imposing the following hard constraints:

1. For every 2D peak, exactly one $X(i, j)$ is true.
2. For every methyl in the structure, at most one $X(i, j)$ is true.
3. For every NOE, exactly one $Y(k, l)$ is true.
4. If $Z(k, k')$ is true, then in the symmetrization graph $k$ and $k'$ must be matched in a maximum matching (MM; outlined below).
5. If $Z(k, k')$ is true, then the methyl assignments of the 2D projections of NOE peaks $k$ and $k'$ must be connected by an appropriate edge in the structure graph, $G$.

In summary, MAUS casts the methyl assignment subgraph isomorphism problem as a system of Boolean constraints captured by a CNF formula and uses CryptoMiniSAT[39], a general-purpose solver that provides either (i) a single graph-matching solution at a millisecond timescale on a single CPU or (ii) a mathematical proof that the formula is unsatisfiable. As there can be multiple mappings consistent with the input constraints, MAUS utilizes the solver to enumerate all valid solutions (see below). In particular, MAUS truncates the solution space of the formula, imposing further restrictions on the range of a node, and then recursively checking for novel solutions in the restricted formula. Due to the fact that it uses a custom-built CNF formula, MAUS has the flexibility to add or remove constraints from the formula and perform 1000s of calculations of the subgraph isomorphism problem in a feasible time frame.

Using these two principles of (1) employing a custom-based formula encoding our system of hard constraints and (2) leveraging a fast and efficient solver, MAUS can perform an exhaustive enumeration of the space of solutions, despite the NP-completeness of the problem, using the following iterative ansatz:

(i) Obtain a single valid mapping of peaks into methyls (validity is imposed by the hard constraints) from the solver.

(ii) Starting from this valid mapping, consider an arbitrary peak (p1) that has been assigned to a particular methyl (m1) in (i) and add a temporary clause to the propositional CNF formula that p1 cannot be assigned to m1.

(iii) Invoke the satisfiability solver a second time to see if it can return an alternative solution for that peak. If it can, then the CNF formula is modified again to include the new solution to the list of excluded assignments and this process is repeated until the solver reaches unsatisfiability (i.e., there are no more assignments of p1 that can lead to a mapping, which satisfies the CNF formula).

(iv) Add hard constraints imposing that p1 can only be assigned to exactly one of the valid options and repeat steps (i) through (iii) to enumerate the assignment options for a different peak, until all peaks have been considered.

(v) Repeat step (iv) until all peaks have been considered.

Through this iterative process, MAUS achieves full enumeration of the peak support sets (containing all valid methyl assignments) in a time that is proportional to the sum of the support set sizes, without resolving to the use of heuristics. If no solutions exist, MAUS must return a proof that all possible assignments have been considered. This method is akin to the celebrated Davis–Putnam–Logemann–Loveland procedure for the satisfiability problem[40,41].

**Reducing complex components of the symmetrization graph**. MAUS reduces the ambiguity arising from spectral overlap in the NOESY data, which is formally described in the form of a bipartite symmetrization graph, allowing it to consider all possible NOE connectivities that can be derived from the raw NOE peaks without any manual effort by the user (Fig. 1d). Simple components of the graph, containing up to two edges, directly impose satisfiability constraints. For each complex component (i.e., a set of nodes containing at least three edges), MAUS first generates all possible MMs using a O($N^3$) algorithm efficiently[42]. Every MM is iteratively examined, introducing additional clauses to the CNF formula and running the satisfiability solver. If MAUS encounters an edge, which causes the formula to become unsatisfiable, then this edge is eliminated from all MMs, repeating until no further edges can be eliminated. Thus, the bipartite graph-matching process is iterated many times as an "outer loop", providing possible constraints to the CNF solver, which serves to eliminate some of the possible

matchings. This allows us to identify a maximal, self-consistent set of upper/lower-diagonal NOEs, which is used to provide the final constraints to the solver, completely removing this burden from the user. During this process, complex components decompose into simple components, recovering additional NOE connectivities into $H$ (Supplementary Table 7). The small fraction of NOE peaks, which remain within complex components, are not used by MAUS.

**Chemical shift-based residue type classifier**. To predict the residue type of a given 2D reference peak when this information is not available from additional experiments, we have analyzed assigned chemical shift data in the BMRB. In particular, we have derived correlated $^{13}C$, $^{1}H$ chemical shift distributions for each methyl atom belonging to Met, Ala, Leu, Val, and Ile, using all assigned resonances in the database. We then constructed a table associating $^{13}C$ and $^{1}H$ pairs with the frequency of each residue type, ranked in decreasing order. A 2D peak can be tentatively assigned to a minimum set of residues, such that their cumulative frequency is ≥99%. Using our benchmark targets, we have evaluated the performance of the classifier for Leu and Val residues, as the resonances of all other residues can be unambiguously determined from their position on two 2D spectra recorded using complementary (MA)ILV and ILV$^{proS}$ samples (Supplementary Table 4).

**Inputs to MAUS**. MAUS accepts as inputs, a user-annotated 2D HMQC list, two 3D NOESY lists recorded using long (300 ms) and short (50 ms) mixing times or a 4D NOESY list recorded using long mixing time (300 ms) and a PDB file. In this work, the experimental peak lists were picked manually using a S/N threshold of 5 and guided by the 2D reference spectrum. When two different users in our group picked the same data sets, they arrived at exactly the same assignment result using MAUS, suggesting that this approach is robust to any biases introduced by the user. This simple process allows us to construct NOE peak tables for each target in under 1 h.

The user can input a crystal structure, an NMR ensemble or an ensemble of models computed using structure prediction methods. MAUS also supports oligomeric systems with regular symmetry, by considering subunit interfaces between different chains in the input PDB file, as specified by the user. Each 2D HMQC peak in the input must be provided in a custom format which specifies residue type(s), a unique number representing the peak, geminal connectivity and stereospecificity information (for peaks corresponding to Leu and Val). To ensure consistency, the user must provide the NMR construct sequence (used to confirm that all methyls present in the NMR samples are also present in the input PDB file) and the labeling scheme used to prepare the NMR sample. The choice of methyl labeling scheme is target-specific and should be optimized such as to achieve the maximum number of probes, while retaining a well-resolved 2D methyl spectrum. Although our manuscript focuses on the most commonly used ILV scheme, MAUS also supports the methyl-bearing side chains of Ala and Met. The user may instruct MAUS to use custom radii (upper limits) for short and long mixing time NOEs (see Table 1 and "Methods" for parameters used for our benchmark targets). The radii must be chosen within recommended ranges (6–8 Å for short-range and 10–12 Å for long-range NOEs). In practice, the smallest possible distance threshold that defines a structure graph that is subgraph isomorphic to the data graph is the optimal distance threshold to run MAUS. A rational approach for choosing this distance is outlined here as follows: (i) we first run MAUS using a standard 8 Å distance threshold for the 50 ms NOESY data, which we assume to be the maximum possible 50 ms NOE distance. This assumption holds for all targets used in this study, also considering that the structure graph already takes into account alternative side-chain rotamers and other local variations of the structure. (ii) We then reduce the threshold by steps of 0.1 until MAUS can no longer fit the NOE network into the structure graph. (iii) The final distance threshold is either the minimum distance identified in ii augmented by 0.5 or 8 Å (we select the smaller value). We find that, for all targets, a threshold in the range 6–8 Å gives optimum results, while always retaining the ground truth within the solution range. This strategy for finding the optimal threshold is carried out manually using a series of short preliminary MAUS runs on our online web-server (in cases where MAUS cannot fit the NOE network, a descriptive message is returned to the user immediately). Running all targets using a conservative 8 Å threshold for the 50 ms NOE data still provides valid and meaningful results, with assignment completeness levels (measured in terms of 1–3 assignment options) that decrease by 3% for HLA-A02, 6% for MBP, 40% for IL-2, and 26% for REC2 targets in our benchmark set, relative to the user-optimized thresholds (Supplementary Table 8). MAUS also employs tunable tolerance parameters, for diagonal NOEs, clustering, and symmetrization, which are set to default values mentioned in previous sections. Detailed instructions for submitting jobs and creating input files in the required format for MAUS are outlined on the help page of our website.

Inconsistencies in peak residue-type information or geminal methyl connectivities in the inputs provided by the user (such as misidentification of a Leu peak as a Val or misidentification of geminal methyls) will, in most cases, lead to an unsatisfiable system of constraints due to inconsistency with the global NOE network. To avoid this and guide the user towards obtaining confident assignments, in the MAUS input page we recommend that the user should only provide residue type and geminal information if they have 100% confidence and are guided by conclusive evidence from a range of additional experimental inputs

(Fig. 2) (residue type and proS-selective labeling, 2D experiments that achieve selective inversion of Leu peaks further assisted by the observation of strong geminal NOEs). We find that, when provided with this information, determination of geminal methyls and residue types can be achieved with 100% confidence by the user as shown by our results on eight benchmark and blind targets. Alternatively, if the user is not confident in the residue type and geminal connectivities, our online interface fully supports a mixed input, where for some methyls this information is either not given or a list of options is provided (e.g., Leu or Val). Finally, the user also has the option to engage the MAUS residue type and geminal methyl classifier, which provides 99% accurate results. Together, these options effectively address any possible inconsistencies in the residue type/geminal methyl user inputs.

Inconsistencies in the input NOE peak lists, due to spectral artifacts, errors in peak picking, which may also include incorrectly identified peak centers. To address this type of data discrepancy, we recommend that users pick 3D or 4D NOESY data at sufficiently high S/N > 5 and remove spectral artifacts manually. However, the most critical form of proactively removing inconsistent data points is provided by our rigorous clustering and symmetrization procedures, which requires that every valid NOE constraint: (i) corresponds to a single methyl group on the 2D $^{13}C/^1H$ reference HMQC spectrum, through the explicit consideration of all possible clustering scenarios, and (ii) arises from two exclusively symmetry-related cross-peaks, through the construction of an explicit bipartite graph between upper- and lower-diagonal NOEs (Fig. 1c, d). The information of which NOE peaks were not considered due to violating either 1 or 2 is provided back to the user in a comprehensive table.

**MAUS output**. MAUS provides a detailed output file with all valid resonance assignment options for each methyl resonance, together with an explanation of how each NOE cross-peak was used by the method. The output file provides information about all non-diagonal NOEs: the 2D peak(s) to which each NOE peak was clustered and its symmetry-related peak. If the NOE peak is not used as a constraint either due to no identifiable symmetric peaks or because it is part of a complex component, which cannot be reduced further, it is indicated in the output file so that the user can revisit the data. In addition, all NOEs corresponding to geminal methyl connectivities are indicated in the output file. This classification summarizes how the NMR data was utilized by MAUS (see Supplementary Table 7 for all benchmark targets used in this study). In the final section, all the resonance assignment options are listed together with a summary of assignment statistics. The total % of resonance assignments for all targets reported here is computed by taking the sum of unambiguous and ambiguous (one to three options) options. If MAUS cannot find a satisfying assignment due to erroneous input data, it provides helpful diagnostic information for the user to troubleshoot their inputs. The input files for all targets in our benchmark set together with detailed description are provided as tutorials in the help page of the MAUS website. MAUS' exhaustive enumeration process, by definition, does not include methyls that are uniquely assigned to one resonance as options in the assignments lists of any other groups with more than one options, due to the use of explicit constraints in the CNF formula, which prevent this from happening. When a single resonance assignment option is provided to the user, this means that MAUS has explored all possible solutions of the graph-matching problem globally and has found that this is the only option that is consistent with the input NOE data graph and any other constraints supplied by the user. This principle provides a powerful tool for deriving highly complete and 100% accurate assignments as follows. For cases with two or three options, the user can manually resolve the correct assignment aided by intensity considerations through inspection of interproton distances in the X-ray or NMR structure (intensities are not considered by MAUS). Here, the user can also consider NOEs that have been ignored by MAUS due to high peak overlap or unclear symmetry, as marked on the annotated output peak table. Methyls for which the correct assignment can be confidently determined by the user can be then fixed in the MAUS input, to trigger a next round of automated assignments. We find that this iterative procedure "propagates" constraints through the data graph, leading to resolution of all assignments in a matter of a few hours while requiring minimal intervention by the user, even for systems with highly complex spectra.

**Transform-restrained Rosetta**. We used the trRosetta protocol, which makes use of the orientation restraints predicted using deep residual neural networks, together with distance restraints from co-evolution information to model structures at a high resolution in the Rosetta force field[25]. We provided as input the sequence of MBP to trRosetta using a web-server available at https://yanglab.nankai.edu.cn/trRosetta/. An ensemble of five models output by trRosetta was supplied to MAUS together with our 2D and 3D NMR data, to obtain the resonance assignments.

**Comparison with publicly available assignment algorithms using NOE peak lists**. Protons were added to X-ray structures using the NIH PDB Utility Server (https://spin.niddk.nih.gov/bax/nmrserver/pdbutil/sa.html). In all simulations, the symmetrization tolerance was set to 0.15, 0.15, and 0.02 p.p.m. for the $^{13}C_{NOE}$, $^{13}C_{indirect}$, and $^1H_{direct}$ dimensions, respectively, except Cas9 HNH domain, for which 0.1 p.p.m. was used for $^{13}C_{NOE}$ and $^{13}C_{indirect}$. In all simulations, the long mixing time NOE distance threshold was set to 10 Å, whereas the short mixing

time NOE distance threshold was optimized for different targets in the 6–8 Å range (see Table 1).

*MAGIC*. The 2D input lists for MAGIC were generated from the NMRFAM-SPARKY-exported 2D $^1H$-$^{13}C$ HMQC list and the 50 ms 3D $C_M$-$C_M$-$H_M$ NOESY list (used to automatically determine geminal pairing). The resulting 2D list was curated for accuracy regarding both methyl residue type and geminal pairing.

*FLAMEnGO*. The experimental chemical shift file (methyl.exp) was generated from the NMRFAM-SPARKY-exported 2D $^1H$-$^{13}C$ HMQC list using an in-house script. Predicted chemical shifts (methyl.pre) were determined using CH3Shift. Line-widths in the 300 ms 3D $C_M$-$C_M$-$H_M$ NOESY data were determined using NMRFAM-SPARKY or CcpNmr. FLAMEnGO v2.4 was used. A total of 100,000 Monte Carlo rounds were performed for each calculation. The final result was taken as the summary of 10 independent runs.

*MAP-XSII*. The experimental chemical shift file (methyl.exp) was generated from the NMRFAM-SPARKY-exported 2D $^1H$-$^{13}C$ HMQC list using an in-house script. Predicted chemical shifts (methyl.pre) were determined using CH3Shift. The weight of chemical shift to NOE was kept as 0.2 (the default). The final result was taken as the summary of ten independent runs.

*MethylFLYA*. The 2D and 3D input peak lists were picked manually in NMRFAM-SPARKY and were used to generate inputs for MethylFLYA[12]. For every target, we carried out 100 independent calculations using three different distance cutoff values (4.5, 5.0, and 5.5 Å) on a single processor to generate expected NOESY peaks from the input structure. Consistent with the parameters used to run all methods, we used $^{13}C$ and $^1H$ chemical shift tolerance values of 0.15 and 0.02 p.p.m., respectively, for all targets with the exception of HNH, where a $^{13}C$ tolerance of 0.1 p.p.m. was used. Consensus results from all runs for assignments classified as strong by MethylFLYA (considered reliable) are reported in Table 2.

**Reporting summary**. Further information on research design is available in the Nature Research Reporting Summary linked to this article.

## Data availability
Nuclear Magnetic Resonance assignments for IL-2, REC2, HNH, and REC3, used as blind targets, have been deposited in the BMRB under accession numbers 28104, 28105, 28106, and 28110, respectively. The unprocessed and processed NMR data along with the structural ensembles and input parameters used to run MAUS are provided for all targets in the help page of the MAUS website. Other data are available from the corresponding author upon reasonable request.

## Code availability
The MAUS server is available at https://methylassignment.chemistry.ucsc.edu.

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

## Acknowledgements
We acknowledge Alison Lindberg and the ITS/ADC staff at UCSC for their assistance in setting up the MAUS website, and Ben Sherman for sharing his python implementation of the satisfiability algorithm designed by Dimitris Achlioptas. This research was supported through NIGMS (R35GM125034) and NIAID (R01AI143997) grants to N.G.S., and through a High-End Instrumentation (HIE) Grant (S10OD018455), which funded the 800 MHz NMR spectrometer at UCSC.

## Author contributions
N.G.S. designed the project. N.G.S and S.N. designed and S.N. implemented the MAUS web-interface. N.G.S., V.S.D.P., and A.C.M. recorded and analyzed NMR experiments. N.G.S., S.N., and V.S.D.P. wrote the paper, with feedback from all authors.

## Competing interests
N.G.S. is listed as a co-inventor in a US Patent (US10871459B2) related to this work. The other authors declare no competing interests.
