## [Peer Review File · Nature Communications]

REVIEWER COMMENTS

Reviewer #1 (Remarks to the Author):

The paper by Nerli, et al describes a protocol for assignment of methyl resonances in NMR spectra without the requirement of preliminary amide backbone assignments. The authors demonstrate the capability of the method, MAUS, using several well-vetted protein systems and other test cases. Overall, I am enthusiastic about the method and find the theoretical development to be rigorous. I believe MAUS is of similar utility to other NMR advancements published in this journal, such as mixed pyruvate labeling reported by Robson, et al, 2018, 9, 356. Given the recent advancements in NMR spectroscopy of large proteins, this method could be quite useful to the broader bio-NMR community. However, there are several points that should be addressed before the manuscript can be considered further.

1) The paper is succinct, but could benefit from a few pieces of additional critical discussion within the text.

First, at what molecular weight does the method break down? There is a good range of protein sizes evaluated in the study, but most, if not all of these examples produce manageable backbone amide NMR spectra, suggesting that the benefit of MAUS in these cases boils down to personal preference rather than necessity. Is there any (even theoretical) limit to the utility of MAUS? It would be useful to comment on how proteins with intractable backbone spectra could be studied with MAUS.

Second, the authors describe a "percent of unique (unambiguous) assignments" with a remaining "percent of resonances with 1-3 possible assignments" while ultimately claiming "100 percent accurate" total spectral assignment. When reading the details of the Supplementary Tables, it becomes clear how the authors achieve and classify nearly 100 percent unambiguous assignments, and that the two pools of resonances don't seem to overlap. However, in the main text, it may be confusing for readers to see how the number of unique identifications and those with 1-3 choices all add up to completely faithful spectral assignments, without explicitly stating summarizing the Supplemental Tables. As written, it could be possible to misinterpret these phrases as an overstating of the results, especially when the statistics are all found in the Supplemental. The authors can address this issue by adding some description of how these groups of resonances evaluated by MAUS produce the final picture.

2) Related to the second point above, are the correct assignment for residues that have multiple (i.e. 1-3) options ever contained within the larger group of singly assigned resonances? That is, if an error in the assignment is made within these two classes of peaks, how would it propagate throughout the results and be detected by MAUS? It becomes clear in the Supplemental that this issue did not appear to be encountered in these test proteins, but the authors may want to discuss this up-front. It may also be useful to comment on the role of the user in making judgments for cases where multiple options to assign a single resonance are available. One could imagine high complexity systems having this issue.

3) The authors mention that proteins without PDB structures can be modeled to ultimately yield "unique and low-ambiguity (2-3 options) assignments for 68 percent of resonances (vs 96 percent with X-ray structures) while maintaining 100 percent accuracy."

How might this percentage be altered with larger and more complex targets than those used here? Some of the most novel and challenging systems often have no X-ray data and can be >40 kDa, and while it is clear that any percentage of spectral assignment for such a protein would be a very useful outcome, the authors may want to speculate further on this point, perhaps in relation to other available methods.

4) The authors should be careful not to overstate the novelty of the HNH nuclease spectral assignments. The backbone and M-I-L-V-A-T methyl resonance assignments have been published

(Biomol. NMR Assign. 2019, 13, 367-370; BMRB entry 27949). At the very least, this paper must be referenced. It may be better, though, to discuss the HNH domain in the context of other comparative test cases in this paper, rather than as a de novo system.

Further, the X-ray structure of HNH used to generate the MAUS data was reported in J. Am. Chem. Soc. 2020, 142, 1348-1358. The authors also note in the Main Text and Supplemental that Cas9 domains maintain identical folds to the full-length protein. Evidence for this was again provided with X-ray and carbon (Ca/Cb/Co) chemical shifts in J. Am. Chem. Soc. 2020, 142, 1348-1358. This paper was not referenced.

5) In the Supplementary Information, the methyl labeling scheme used for each protein appears slightly different. Is MAUS expected to perform equally well if full Met-Ile-Leu-Val-Ala-Thr methyl resonances were present in the spectrum?

Reviewer #2 (Remarks to the Author):

The manuscript by Nerli et al. reports a new workflow to assign methyl resonances from a limited set of NMR data and a known 3D structure of the target protein. Methyl groups are important reporters for the study of large protein or complexes by NMR. However, assigning NMR resonances frequencies to methyl groups is a challenging task when using traditional approaches that involve either the acquisition, processing and analysis of numerous 3D or 4D spectra or site-directed mutations of many (if not all) amino acids positions with methyl-bearing side-chains.

In this context, any fully automated or partially automated technique to assign methyl resonances can be considered as an advance in the field. As of today, less than an handful of software can perform this task with sufficient accuracy. Their common denominator is the necessity to have a 3D structure of the protein (or complex) and the use of methyl-methyl NOESY data.

The proposed approach, names MAUS, does not requires previous assignments of backbone resonances. However, a known 3D structure of the protein is mandatory. Considering the typical usage of methyl resonances in structural biology by NMR, it cannot be considered as a limitation of the approach. In terms of NMR data, the minimal input for MAUS consist of an annotated 2D ¹³C-¹H spectrum and two CCH NOESY spectra (or a single 4D HCCH spectrum). To be more precise, MAUS uses as input lists of peaks (i.e. the outcome processing and manual curation of raw spectra).

The algorithmic process of MAUS involves the construct of a graph from provided structure on one side and of smaller graphs from NOE data after clustering to the 2D peaks using the matched frequencies. Then a subset of data graphs is found by fitting to the structure graph. Since the technical subtleties behind the algorithm uses to solved the problem are behind my scope of expertise, I will not review this part of the manuscript.

The approach was benchmarked on 8 proteins (4 for which reference assignments where taken from the BMRB database and 4 more targets treated as de novo problems). In all cases the necessary NMR data were recorded by the authors. For the 4 de novo cases, MAUS assignment were validated using traditional methods using backbone and side-chains assignments. MAUS is reported to provide unique assignments for 59% to 89% of the methyl resonance, depending on the target. In all cases, 100% of solutions given by MAUS correspond (or contain) to the correct assignment. For 2 targets (HLA-A002 and REC2) , a non-trivial number of methyls have more than 3 assignment options. The efficiency of the software is compared to 3 others methods, namely MAGIC, FLAMEnGO and MAP-XSII.

The manuscript is well-written but rather compact and dense which make is sometimes difficult to follow, especially since there is no explicit structuration. The work is of high-quality, both experimentally and computationally and certainly relevant for the NMR community.

However, two major points need to be addressed before possible publication of the manuscript:

1) Although it is cited in the introduction, no comparison is made with the very recent methylFLYA approach from the Guntert's group (Pritišanac et al, Nat Comm, Oct 2019). methylFLYA has exactly the same purpose as MAUS with very similar user inputs. According to the Pritišanac' publication, it exceeds all other software (including MAGIC, FLAMEnGO and MAP-XSII) in term of accuracy.

The authors must provide a comparison with methylFLYA on the 8 targets used in their benchmark. If the comparison is not possible, it must be justified and stated in the manuscript.

2) Tables 1 gives "optimized" distance threshold (for 50ms NOESY) for the different tested targets. I guess the optimization means "to find the distance threshold yielding the best accuracy". If the distance has to be optimize for each target, it means it has a significant role in the efficiency of the algorithm.

To convince the reviewer (and the readership) of general applicability of the software, the authors must provide results for MAUS using an objectively (or rationally) chosen distance threshold.

Some minor points that would help improve the manuscript are also listed below.:

- Supplementary table 4 would deserve to be a main result since it is the only item where the accuracy of MAUS is explicitly reported.

- Owing to the nature of the algorithm (I suspect) no score or probability or ranking is given to the proposed solutions. Can the authors discuss on ways for the user to assess the quality of the proposed assignments (especially when more than one solution are given).

- Out of curiosity, I tried to run MAUS without any annotation of the 2D peaks (i.e. giving all possible residue types from the labeling scheme). As expected the accuracy rate dropped substantially. While this may not reflect a "real-life" case (an expert user would provide precise annotation that are readily accessible), it could be interesting to get the statistics for all tested targets when no relevant user-annotation are provided to the 2D HMQC peak-list.

- End of paragraph 4, page 3, in the sentence "This approach relieves the user from the burden of interpreting raw data, through an explicit and exhaustive consideration of all possible data graphs consistent with the input NOE peaks". What do the authors mean by « raw data » ?

- Paragraph 8, page 4. The sentence "The structures of targets which lack a representative template in the PDB can be modeled using sequence co-evolution approaches^{19,20}" appears a bit out of the blue.

I have the impression the author went one step ahead too quickly. When no structure is known, first intention would be to resort to comparative modeling and try to get a model from an homologous protein for witch the structure is known. If no template exists, then co-evolution approach can provide a mean to obtain an acceptable 3D model. I thus invite the authors to describe this strategy more clearly in the manuscript.

- Same paragraph: "Since structure prediction methods may also lead to globally incorrect models, we also demonstrated that MAUS can distinguish between related protein folds directly from unprocessed NOE peaks. »

What do the authors mean by « unprocessed NOE peaks » ?

- Last paragraph, page 5 : « In summary, we demonstrate that a satisfiability-based approach can deliver reliable assignments for a range of targets amenable to solution-state NMR, using unprocessed NOE peak lists and an existing structural model with the correct overall fold. »
Again, what do the authors mean by « unprocessed NOE peak lists » ?

- The server to run MAUS is functional with the data provided from the help page. However, I never got it to work with a 4D HCCH peak list in Sparky format. I do not blame the authors for that, it can certainly be that I did not provide data in the expected format. Yet, the error message obtained from the server upon failure is generic and, while it suggests how to rerun, it does not mention the actual source of error (format, incorrect sequence etc).

- In the method section, subsection "Generating a structure graph, G". The equation to obtain an average methyl-methyl distance uses a "r-6-averaging" (i.e. with multiplicity correction) instead of the more commonly used "r-6 summation". Can the authors justify the use of "r-6-averaging" ?

Reviewer #3 (Remarks to the Author):

Major revisions

The article of Nerli & De Paula et al. presents an interesting study about automated methyl group resonances assignment of protein by solution-state NMR spectroscopy. As the authors mentioned in the introduction, this is an active field of research with several recent contributions (Monneau et al. (2017), Xu et al. (2013), Chao et al. (2014), Pristisana et al. (2019)). The ultimate goal is crucial because large proteins cannot be studied with conventional NMR assignment approaches. Therefore, only methods relying on sparse NMR restraints can overcome this bottleneck. The presented approach would be to the benefit of the whole structural biology community. The authors claim to outperform the currently available methods, which would be a significant step forward. But some important points are missing. I strongly recommend to address these points before publication:

MAUS Robustness:

1. "MAUS uses the NOE network together with additional experimental inputs, such as peak residue type information and geminal methyl resonance connectivities, to build a system of hard constraints."

How MAUS performs if the experimental input data contain inconsistencies?

2. Were the experimental peak lists manually curated / idealized? The peak lists should not be cleaned by the authors. The programs ccpnmr, sparky, or nmrDraw can automatically pick peaks. How MAUS behave with artifacts in the input peak files?

MAUS Solver:

3. "MAUS leverages a special-purpose constraint satisfaction solver"

This is the central part of the methodology, and it is not really described. I understand that the mathematical description of the solver is beyond the scope of the publication, but a short description should be included.

4. It is not clear if MAUS will always find all exact solutions to the NP-complete problem or if the algorithm is partially heuristic. This should be clearly stated.

5. Explain how MAUS can find the exact solutions of an NP-complete problem in a short time. To me, either MAUS is partially heuristic or finding the solution should be "slow". Please explain that point in the manuscript.

Other comments:

6. "performance of MAUS on deriving de novo, blind assignments
The assignment cannot be de novo blind if the authors know the assignments.

7. "Our results using a benchmark set with known assignments and several blind targets show that, unlike previous methods, MAUS provides 100% accurate solutions thereby greatly reducing manual effort, costs and errors introduced due to human pre-processing of the data."
The program MAGMA also provided 100% accurate solutions; please rephrase and or comment.

8. "Other publicly available methyl assignment methods (MAGIC/FLAMEno2.4/MAP-XSII) operating on the same inputs had a significant error rate of up to (34/16/56%)."
Please also compare to methylFLYA. methylFLYA outperforms MAGIC/FLAMEno2.4/MAP-XSII.

9. "To circumvent the need for existing backbone assignments, methyl NOE data and a known structure of the target protein are used to derive a set of possible assignments, by fitting local networks of NOEs to methyl distances derived from the 3D structure..."
The mentioned methods also use long-range NOEs. Explain why you call it "local" or modify the statement.

10. "exhaustive mapping of the global NOE network to the target structure. This strategy was first outlined in the method MAGMA" ... "enumerate all assignments which satisfy the maximum number of NOE connectivities."

Similar approaches were employed in a different context. I suggest the authors to read the article of Constantine et al. as well as Orts et al. (see below).

The first reference also covers bipartite graph matching problem, while the second uses sparse NOE data and also enumerates all assignments compatible with the NOE data set.

K. L. Constantine, M. E. Davis, W. J. Metzler, L. Mueller and B. L. Clau (2006) J Am Chem Soc 128 22 7252-63

Protein-ligand NOE matching: a high-throughput method for binding pose evaluation that does not require protein NMR resonance assignments

J. Orts, M. A. Walti, M. Marsh, L. Vera, A. D. Gossert, P. Guntert and R. Riek (2016) Journal of the American Chemical Society 138 13 4393-4400

NMR-Based Determination of the 3D Structure of the Ligand-Protein Interaction Site without Protein Resonance Assignment

11. Figure S1.

Define the "effective degree connectivity"

12. Table S1

13. Please double-check this table. The % of unique assignment are the same between different methods and the time does not support your conclusion (MAGMA is faster than MAUS).

14. Please describe Table S5.

REVIEWER COMMENTS

Reviewer #1 (Remarks to the Author):

The paper by Nerli, et al describes a protocol for assignment of methyl resonances in NMR spectra without the requirement of preliminary amide backbone assignments. The authors demonstrate the capability of the method, MAUS, using several well-vetted protein systems and other test cases. Overall, I am enthusiastic about the method and find the theoretical development to be rigorous. I believe MAUS is of similar utility to other NMR advancements published in this journal, such as mixed pyruvate labeling reported by Robson, et al, 2018, 9, 356. Given the recent advancements in NMR spectroscopy of large proteins, this method could be quite useful to the broader bio-NMR community. However, there are several points that should be addressed before the manuscript can be considered further.

We thank the reviewer for their positive comments on the practical utility of our method. We provide a detailed response to the points raised by the reviewer below, with revisions highlighted in yellow in our manuscript.

1) The paper is succinct, but could benefit from a few pieces of additional critical discussion within the text.

Additional critical discussion points have been added in the revised version of our manuscript, as outlined here.

First, at what molecular weight does the method break down? There is a good range of protein sizes evaluated in the study, but most, if not all of these examples produce manageable backbone amide NMR spectra, suggesting that the benefit of MAUS in these cases boils down to personal preference rather than necessity. Is there any (even theoretical) limit to the utility of MAUS? It would be useful to comment on how proteins with intractable backbone spectra could be studied with MAUS.

The reviewer raises an important point regarding the size limitation of MAUS. Methyl-based NMR assisted by MAUS assignments, can address proteins well above the current backbone amide-based solution NMR size limit, with the main limiting factor being the quality of the input methyl NOE data. While most proteins in our benchmark set can -in principle- provide backbone spectra of sufficient quality to carry out near-complete assignments using multiple 3D and 4D triple-resonance experiments and a significant time investment in analyzing the data by an expert user, in practice this can be further challenged by solvent exchange of the amide protons (which also limits the pH range that can be used to study these proteins) and conformational exchange-induced line broadening, due to intrinsic dynamic processes. Both of these limiting factors necessitate extensive sample optimization to be carried out prior to backbone-based NMR studies. To demonstrate this point, we have provided a real-life example of a protein with an intractable backbone amide spectrum. The Cas9 REC3 domain shows a significant fraction of completely broadened amide resonances (~55%, residues 660-712), most likely due to a slow exchange conformational process under our sample conditions, however, with the use of methyl probes and our MAUS workflow we can readily obtain complete methyl assignments in a matter of a few days.

An additional point worth considering is that, in order to obtain the maximum gains from the TROSY effect and to improve ^{13}C relaxation for carrying out triple-resonance experiments, extensive deuteration of the $\text{H}\alpha$ and aliphatic protons is needed for proteins above 25 kDa. Protons are then re-introduced at the amide positions, often with the use of *in vitro* refolding protocols. However, in many cases of multi-domain proteins with complex, disulfide-bonded topologies, refolding protocols are not applicable, and as a result the amide resonances corresponding to the structural core and require long exchange times are absent for many biologically important residues. These limitations can be overcome by switching to selectively labeled methyl groups with the added benefit of a large increase in S/N, relative to the amide resonances. For these reasons, we believe that MAUS is a practically useful tool to study not only larger systems that are intractable by backbone-based NMR, but also enabling routine applications for the study of medium-range targets by completely circumventing the need to obtain backbone assignments in a preceding step.

To test the size limit supported by MAUS, as suggested by the reviewer, we expanded our simulation benchmark set to include 63 PDB structures with structure graphs containing more than 100 methyls from Ile, Leu and Val residues (corresponding to system sizes of up to 116 kDa) (**Supplementary Fig. 5**). We chose to simulate assignment cases corresponding to the ILV (as opposed to AILV) labeling scheme, since a large fraction of the resonances corresponding to Ala methyls are missing from the spectra of larger proteins. For each target, a data graph was simulated by removing edges from the corresponding structure graph (defined as all methyl connectivities up to 10 Å present in the 3D structure) until a degree connectivity (defined as $2 * \text{number of edges} / \text{number of nodes}$) of 3.5 or higher was reached. This value was chosen to represent experimental cases with 1.75 – 2.1 observed NOEs/methyl resonance, which are characteristic of real-life data sets. We find that, as long as this requirement is satisfied, MAUS can efficiently tackle

the computational complexity of the graph matching problem and provide meaningful assignments in a reasonable time, even for larger targets (up to 4 h on a single CPU). Specifically, our simulation results show that we can obtain a high coverage (60-80%) of methyl groups with 1-3 residue assignment options for targets up to 352 methyls. Moreover, MAUS can address proteins of up to 716 methyls/200 kDa (as exemplified by the Teneurin 2 partial extracellular domain, PDB ID: 6FB3), albeit with a longer run time (11 h), which is still feasible from a computational standpoint. This simulation test demonstrates that selective methyl labelling, supported by MAUS, can render biological systems that are intractable by amide-based methods amenable to solution-based NMR studies. We have clarified these points in the revised manuscript on pages 4, and 5 and added a new Supplementary Figure 5 to highlight the new simulation results.

Supplementary Figure 5. Performance of MAUS on simulated ILV data graphs generated from targets of sizes up to 116 kDa. (a) Target size distribution (in terms of number of methyl groups from Ile, Leu and Val residues) of 63 proteins with high-resolution PDB structures used in this simulation (top). For each target, a data graph was simulated by removing edges from the corresponding structure graph (defined as all methyl connectivities up to 10 Å present in the 3D structure) until a degree connectivity ($2 \times \text{number of edges} / \text{number of nodes}$) of 3.5 or higher is reached. A bar plot showing the degree connectivity of all simulated data graphs from these structures (bottom), representing cases with 1.75 – 2.1 observed NOEs/methyl group that are characteristic of real-life experimental data sets. (b) Scatter plot showing run time (in minutes) taken by MAUS to assign resonances for 63 targets of size up to 116 kDa (352 methyls; PDB ID: 5WTI). Colors indicate % of methyls with up to 3 assignment options, according to the scale on the right.

Second, the authors describe a “percent of unique (unambiguous) assignments” with a remaining “percent of resonances with 1-3 possible assignments” while ultimately claiming “100 percent accurate” total spectral assignment. When reading the details of the Supplementary Tables, it becomes clear how the authors achieve and classify nearly 100 percent unambiguous assignments, and that the two pools of resonances don't seem to overlap. However, in the main text, it may be confusing for readers to see how the number of unique identifications and those with 1-3 choices all add up to completely faithful spectral assignments, without explicitly stating summarizing the Supplemental Tables. As written, it could be possible to misinterpret these phrases as an overstating of the results, especially when the statistics are all found in the Supplemental. The authors can address this issue by adding some description of how these groups of resonances evaluated by MAUS produce the final picture.

We apologize for not being sufficiently clear in the presentation of our results. As suggested by the reviewer we have added a statement in the “Methods” section (“MAUS output” subsection) describing how we compute assignment statistics, and we have also moved Supplementary Table 4 to Table 2 in order to make it explicitly clear how the number of unique assignments and those with 1-3 options add up.

2) Related to the second point above, are the correct assignment for residues that have multiple (i.e. 1-3) options ever contained within the larger group of singly assigned resonances? That is, if an error in the assignment is made within these two classes of peaks, how would it propagate throughout the results and be detected by MAUS? It becomes clear in the Supplemental that this issue did not appear to be encountered in these test proteins, but the authors may want to discuss this up-front. It may also be useful to comment on the role of the user in making judgments for cases where multiple options to assign a single resonance are available. One could imagine high complexity systems having this issue.

This is an important point raised by the reviewer, which further highlights an important feature of our method: MAUS' exhaustive enumeration process, by definition, does not include methyls that are uniquely assigned to one resonance as options in the assignments lists of any other groups with more than one options, due to the use of explicit constraints in the satisfiability formula which prevent this from happening. When a single resonance assignment option is provided to the user, this means that MAUS has explored all possible solutions of the graph matching problem globally, and has found that this is the only option that is consistent with the input NOE data graph and any other constraints supplied by the user. This principle provides a powerful tool for deriving highly complete and 100% accurate assignments, as follows. For cases with 2 or 3 options, the user can manually resolve the correct assignment aided by intensity considerations through inspection of inter-proton distances in the X-ray or NMR structure (intensities are not considered by MAUS). Here, the user can also consider NOEs that have been ignored by MAUS due to high peak overlap or unclear symmetry, as marked on the annotated output peak table. Methyls for which the correct assignment can be confidently determined by the user can be then fixed in the MAUS input, to trigger a next round of automated assignments. We find that this iterative procedure "propagates" constraints through the data graph, leading to resolution of all assignments in a matter of a few hours while requiring minimal intervention by the user, even for systems with highly complex spectra. We have clarified this important point under the "MAUS output" subsection in the "Methods" section on pages 26 of the revised manuscript.

3) The authors mention that proteins without PDB structures can be modeled to ultimately yield "unique and low-ambiguity (2-3 options) assignments for 68 percent of resonances (vs 96 percent with X-ray structures) while maintaining 100 percent accuracy."

How might this percentage be altered with larger and more complex targets than those used here? Some of the most novel and challenging systems often have no X-ray data and can be >40 kDa, and while it is clear that any percentage of spectral assignment for such a protein would be a very useful outcome, the authors may want to speculate further on this point, perhaps in relation to other available methods.

We share the reviewer's concerns. In the absence of X-ray or NMR structures, a model can be computed using state-of-the-art methods such as trRosetta which can yield high precision and accuracy models for multi-domain protein targets in the CASP (Critical Assessment of Structure Prediction) set. Alternatively, the user may apply a divide and conquer strategy as shown in our current paper for Cas9 and by many other groups studying other important multi-domain proteins. In both cases, using a structural model that is accurate within 2-3 Å and 3D or 4D NOESY data picked at sufficiently high signal to noise (S/N > 5) ratio levels, MAUS can provide meaningful assignment options for a large fraction (60-80%) of methyl groups. We have added this clarification in the revised manuscript on page 6.

4) The authors should be careful not to overstate the novelty of the HNH nuclease spectral assignments. The backbone and M-I-L-V-A-T methyl resonance assignments have been published (Biomol. NMR Assign. 2019, 13, 367-370; BMRB entry 27949). At the very least, this paper must be referenced. It may be better, though, to discuss the HNH domain in the context of other comparative test cases in this paper, rather than as a *de novo* system.

Further, the X-ray structure of HNH used to generate the MAUS data was reported in J. Am. Chem. Soc. 2020, 142, 1348-1358. The authors also note in the Main Text and Supplemental that Cas9 domains maintain identical folds to the full-length protein. Evidence for this was again provided with X-ray and carbon (Ca/Cb/Co) chemical shifts in J. Am. Chem. Soc. 2020, 142, 1348-1358. This paper was not referenced.

We thank the reviewer for pointing this important recent work, and we have now referenced both articles on page 5 of the revised manuscript. We have considered the HNH nuclease as a *de novo* target since the backbone and MILVAT methyl resonances were published concurrently with the development of our work, and the published assignments were not available while running MAUS. It is worth noting that our automated, backbone independent methyl assignments are in 100% agreement with the published methyl assignments which were derived through the more laborious, backbone-based approach, which further highlights the practical utility of our work in saving machine time and manual effort spent.

5) In the Supplementary Information, the methyl labeling scheme used for each protein appears slightly different. Is MAUS expected to perform equally well if full Met-Ile-Leu-Val-Ala-Thr methyl resonances were present in the spectrum?

The choice of methyl labeling scheme is target-specific, and should be optimized such as to achieve the maximum number of probes while retaining a well-resolved 2D methyl spectrum. Although our manuscript focuses on the most commonly used ILV scheme, the current online version of MAUS also supports the methyl-bearing side chains of alanine and methionine. The threonine option is not provided currently on the web-interface but, can be added at a

later stage based on feedback from the NMR community. We have added these statements explaining the choice of labeling schemes under the “*Input to MAUS*” section on page 24 of the revised manuscript.

Reviewer #2 (Remarks to the Author):

The manuscript by Nerli et al. reports a new workflow to assign methyl resonances from a limited set of NMR data and a known 3D structure of the target protein. Methyl groups are important reporters for the study of large protein or complexes by NMR. However, assigning NMR resonances frequencies to methyl groups is a challenging task when using traditional approaches that involve either the acquisition, processing and analysis of numerous 3D or 4D spectra or site-directed mutations of many (if not all) amino acids positions with methyl-bearing side-chains.

In this context, any fully automated or partially automated technique to assign methyl resonances can be considered as an advance in the field. As of today, less than an handful of software can perform this task with sufficient accuracy. Their common denominator is the necessity to have a 3D structure of the protein (or complex) and the use of methyl-methyl NOESY data.

The proposed approach, names MAUS, does not requires previous assignments of backbone resonances. However, a known 3D structure of the protein is mandatory. Considering the typical usage of methyl resonances in structural biology by NMR, it cannot be considered as a limitation of the approach. In terms of NMR data, the minimal input for MAUS consist of an annotated 2D ¹³C-¹H spectrum and two CCH NOESY spectra (or a single 4D HCCH spectrum). To be more precise, MAUS uses as input lists of peaks (i.e. the outcome processing and manual curation of raw spectra).

The algorithmic process of MAUS involves the construct of a graph from provided structure on one side and of smaller graphs from NOE data after clustering to the 2D peaks using the matched frequencies. Then a subset of data graphs is found by fitting to the structure graph. Since the technical subtleties behind the algorithm uses to solved the problem are behind my scope of expertise, I will not review this part of the manuscript.

The approach was benchmarked on 8 proteins (4 for which reference assignments where taken from the BMRB database and 4 more targets treated as de novo problems). In all cases the necessary NMR data were recorded by the authors. For the 4 de novo cases, MAUS assignment were validated using traditional methods using backbone and side-chains assignments. MAUS is reported to provide unique assignments for 59% to 89% of the methyl resonance, depending on the target. In all cases, 100% of solutions given by MAUS correspond (or contain) to the correct assignment. For 2 targets (HLA-A002 and REC2), a non-trivial number of methyls have more than 3 assignment options. The efficiency of the software is compared to 3 others methods, namely MAGIC, FLAMEnGO and MAP-XSII.

The manuscript is well-written but rather compact and dense which make is sometimes difficult to follow, especially since there is no explicit structuration. The work is of high-quality, both experimentally and computationally and certainly relevant for the NMR community.

We thank the reviewer for their positive comments on the quality of our work. As outlined in our point-by-point response, we have incorporated additional results and explanatory sections in the revised manuscript (highlighted in yellow) to make it easier to understand by the general structure-based NMR community.

However, two major points need to be addressed before possible publication of the manuscript:

1) Although it is cited in the introduction, no comparison is made with the very recent methylFLYA approach from the Guntert’s group (Pritišanac et al, Nat Comm, Oct 2019). methylFLYA has exactly the same purpose as MAUS with very similar user inputs. According to the Pritišanac’ publication, it exceeds all other software (including MAGIC, FLAMEnGO and MAP-XSII) in term of accuracy.

The authors must provide a comparison with methylFLYA on the 8 targets used in their benchmark. If the comparison is not possible, it must be justified and stated in the manuscript.

We thank the reviewer for bringing up this important point. The main difference with MAUS, based on the information provided in the manuscript by Pritišanac et al, is that MethylFLYA is a non-exhaustive method, which means that it can -in principle- provide wrong assignments to the user. The main differences between MAUS and other methods from a practical standpoint are provided in the table below, which is also included in the revised manuscript as a new Supplementary Table 5.

Supplementary Table 5: Comparison of MAUS with existing methyl assignment methods

Method	Availability (academic users)	May output erroneous assignments	Provides >50% unambiguous assignments	Runtime in min (worst case)	Runs directly on 3D/4D NOE peak data
MethylFLYA	Commercial	Yes	No	~2,553	Yes
MAGMA	Free	No	Yes	Time out ^{&}	No
MAGIC	Free	Yes	No	Time out [*]	Yes
FLAMEnGO2.4	Free	Yes	No	Time out [*]	Yes
MAP-XSII	Free	Yes	No	~16	Yes
MAUS	Free (Web-server)	No	Yes	~16 [%]	Yes

[&] After 167 min of runtime (MAGMA limit)

^{*} After 96 hr of runtime

[%] Includes time taken by MAUS to run the *Rosetta* side chain optimization protocol, evaluate all possible data graphs consistent with the data, and exhaustively enumerate all assignment solutions

The reason we did not attempt a side-by-side comparison on our benchmark set of 8 targets in the original version of our paper is that, as opposed to MAUS that is readily available to the NMR community, MethylFLYA must be obtained via a commercial software license and requires a working knowledge of the INCLAN programming language by the user. Following the reviewer's suggestion, we have purchased a license from the authors and carried out a detailed comparison of MAUS with MethylFLYA on the eight benchmark targets used in our study, by providing the identical inputs to run both methods. Upon comparison of the number of resonances assigned uniquely by MAUS and with high confidence by MethylFLYA, we find that MAUS outperforms MethylFLYA for 6/8 targets in our set (H β_2 m, MBP, IL-2, HNH, REC2 and REC3). Specifically, (i) for IL-2 and REC2, MethylFLYA reports a significantly smaller number of confident assignments (38% and 10%) relative to MAUS (74% and 60%), which includes 2 resonances that are incorrectly assigned by MethylFLYA in each target, (ii) for HNH and MBP, it produces confident assignments at completeness levels that are 6 and 7% lower relative to MAUS, respectively and (iii) for H β_2 m and REC3, MethylFLYA crashes (segmentation fault), producing no results. While MethylFLYA assigns a higher % (11 & 17%) of methyl resonances for two targets in our set (HLA-A01 & HLA-A02), it takes a prohibitively longer computation time (11 & 17 hr vs 15 & 17 min for MAUS) on a single CPU, and in general MethylFLYA is 2-3 orders of magnitude less efficient than MAUS (Supplementary Table 6). Therefore, in order for users to obtain results using MethylFLYA in a reasonable time they must also have access to a local CPU cluster, further limiting broad adoption of the method, while MAUS can be accessed by users through an online web-interface. Finally, it is worth noting that, while MethylFLYA additionally provides low-confidence assignments for some resonances, MAUS provides a list of options for all methyl resonances which is guaranteed to contain the correct assignment, due to the fact that it is using a satisfiability-based enumeration process that is both exact and exhaustive. This comparison has been included in pages 5 and 27 of the revised text and Supplementary Tables 5 and 6.

Supplementary Table 6: Comparison of MAUS and MethylFLYA on our benchmark set of 8 targets

Target	# of methyls	# of unique assignments [%]		Time (in minutes)	
		MAUS	MethylFLYA	MAUS	MethylFLYA
H β_2 m (AILV)	35	31/0	N/D [*]	7	N/D [*]
HLA-A01 (AILV)	87	56/0	64/0	15	704
HLA-A02 (AILV)	94	66/0	77/0	17	1024
MBP (MILVproS)	76	57/0	52/0	19	853
IL-2 (ILV)	61	45/0	23/2	5	2553
HNH (ILV)	53	47/0	44/0	4	814
REC2 (ILV)	69	41/0	7/2	5	1653
REC3 (ILV)	85	57/0	N/D [*]	14	N/D [*]

N/D^{*}- (Not determined) MethylFLYA returned an error

[%]- # of methyls with 1 option correct / # of methyls wrongly assigned

2) Tables 1 gives "optimized" distance threshold (for 50ms NOESY) for the different tested targets. I guess the optimization means "to find the distance threshold yielding the best accuracy". If the distance has to be optimized for each target, it means it has a significant role in the efficiency of the algorithm.

To convince the reviewer (and the readership) of general applicability of the software, the authors must provide results for MAUS using an objectively (or rationally) chosen distance threshold.

The reviewer is understandably concerned about the choice of the 50 ms NOE distance threshold in a general setting. In practice, the smallest possible distance threshold that defines a structure graph that is subgraph-isomorphic to the data graph is the optimal distance threshold to run MAUS. A rational approach for choosing this distance is outlined here: *i)* We first run MAUS using a standard 8 Å distance threshold for the 50 ms NOESY data, which we assume to be the maximum possible 50 ms NOE distance. This assumption holds for all targets used in this study, also considering that the structure graph already takes into account alternative sidechain rotamers and other local variations of the structure. *ii)* We then reduce the threshold by steps of 0.1 until MAUS can no longer fit the NOE network into the structure graph. *iii)* The final distance threshold is either the minimum distance identified in *ii)* augmented by 0.5 Å, or 8 Å (we select the smaller value). We find that, for all targets, a threshold in the range 6-8 Å gives optimum results, while always retaining the ground truth within the solution range. Running all targets using a conservative 8 Å threshold for the 50 ms NOE data still provides meaningful results, with assignment completeness levels (measured in terms of 1-3 assignment options) that drop by 3% for HLA-A02, 6% for MBP, 40% for IL-2 and 26% for REC2 targets in our benchmark set. We have added a new Supplementary Table 8 showing the performance of MAUS on the benchmark targets using 8 Å 50 ms NOE distance threshold.

Supplementary Table 8: Performance of MAUS using 8 Å 50ms NOE distance threshold

Target	% unique assignments	% options <= 3 and > 1
Hβ ₂ m	77	23
HLA-A01	64	30
HLA-A02	65	17
MBP	61	29
IL-2	26	31
HNH	89	11
REC2	33	12
REC3	67	28

We apologize that this rational approach for choosing the distance threshold was not clear in the original manuscript, and have added this clarification in the revised manuscript under the section “*Input to MAUS*” on page 24.

Some minor points that would help improve the manuscript are also listed below.:

- Supplementary table 4 would deserve to be a main result since it is the only item where the accuracy of MAUS is explicitly reported.

We have moved the Supplementary Table 4 (now Table 2) to the “*main results*” section in the revised manuscript.

- Owing to the nature of the algorithm (I suspect) no score or probability or ranking is given to the proposed solutions. Can the authors discuss on ways for the user to assess the quality of the proposed assignments (especially when more than one solution are given).

The reviewer is correct, no score or ranking is given to the proposed solutions with two or more options. While the different options can be further ranked using sampling from the theory of exact computing, an avenue that is currently pursued in our group, this requires significant additional code development which falls outside the scope of our current paper.

We find that both in simulations and our real-life data, the proposed assignments with more than one solution always contain the correct option, narrowing down the possibilities that are presented to the user for manual disambiguation using an iterative process: For cases with 2 or 3 options, the user can manually resolve the correct assignment aided by intensity considerations through inspection of inter-proton distances in the X-ray or NMR structure (intensities are not considered by MAUS). Here, the user can also consider NOEs that have been ignored by MAUS due to high peak overlap or unclear symmetry, as marked on the annotated output peak table. Methyls for which the correct assignment can be confidently judged by the user can be then fixed in the MAUS input, to trigger a next round of automated assignments. We find that this iterative procedure “propagates” constraints through the data graph, leading to resolution of all assignments in a matter of a few hours while requiring minimal intervention by the user, even for systems with highly complex spectra. We have added this clarification in the “*MAUS Output*” section on page 26 of the revised manuscript.

- Out of curiosity, I tried to run MAUS without any annotation of the 2D peaks (i.e. giving all possible residue types from the labeling scheme). As expected, the accuracy rate dropped substantially. While this may not reflect a “real-life” case (an expert user would provide precise annotation that are readily accessible), it could be interesting to get the statistics for all tested targets when no relevant user-annotation are provided to the 2D HMQC peak-list.

We have provided a table in the Supplementary information (Supplementary Table 4) that detail MAUS accuracy when the annotations of Leu and Val are removed. We showed in the manuscript that introducing ambiguity between Leu/Val methyl resonances, led to 18% less uniquely assigned peaks, however, the accuracy of all assignment options remained 100%.

If we do not have user annotations for residue labels, we inherently lose information about stereospecificity (or geminals in the case of leucine and valine). Following the reviewer’s suggestion, we ran MAUS without any annotation of the 2D peaks for the proS labeled MBP target protein. In this case, MAUS is still able to successfully assign 71% of resonances uniquely and 17% of resonances with 2 or 3 options with 100% accuracy. We have now included this result on page 4 of the revised manuscript.

- End of paragraph 4, page 3, in the sentence “This approach relieves the user from the burden of interpreting raw data, through an explicit and exhaustive consideration of all possible data graphs consistent with the input NOE peaks”. What do the authors mean by « raw data »?

By raw data we mean the NOE peaks (CCH or CHCH coordinates) identified in the NOESY spectra. Specifically, while MAGMA requires a data graph to be constructed by the user by manual inspection of the 3D or 4D NOESY peaks, MAUS utilizes the unprocessed NOE peak lists to construct all possible data graphs via separate clustering and symmetrization processes. Thus, MAUS treats the data graph generation process as a satisfiability problem in and of itself, and uses a massive amount of computation to explicitly consider all possible ways of interpreting the raw NOE peaks into distance constraints, thereby removing this burden from the user. Here, unprocessed or raw data refers to peaks picked (or peak lists) from 3D or 4D NOESY data. We have clarified this point on pages 3 and 6 of the revised manuscript.

- Paragraph 8, page 4. The sentence “The structures of targets which lack a representative template in the PDB can be modeled using sequence co-evolution approaches^{19,20}” appears a bit out of the blue. I have the impression the author went one step ahead too quickly. When no structure is known, first intention would be to resort to comparative modeling and try to get a model from a homologous protein for which the structure is known. If no template exists, then co-evolution approach can provide a mean to obtain an acceptable 3D model. I thus invite the authors to describe this strategy more clearly in the manuscript.

We agree with the reviewer’s choice of strategy for modeling the structures of unknown targets using homology-based modelling first, and have incorporated this as a suggested route in the manuscript. As the reviewer pointed out, the first step when X-ray structures from homologous sequences are unavailable should be to perform comparative modeling, using standard protocols such as RosettaCM (Song et al., 2013). Only in the absence of X-ray structure and a high-confidence homology-based model, we recommend using sequence co-evolution approaches such as trRosetta (Yang et al., 2020). We have clarified this point on page 6 of the revised manuscript.

- Same paragraph: “Since structure prediction methods may also lead to globally incorrect models, we also demonstrated that MAUS can distinguish between related protein folds directly from unprocessed NOE peaks. » What do the authors mean by « unprocessed NOE peaks »?

- Last paragraph, page 5: « In summary, we demonstrate that a satisfiability-based approach can deliver reliable assignments for a range of targets amenable to solution-state NMR, using unprocessed NOE peak lists and an existing structural model with the correct overall fold. »
Again, what do the authors mean by « unprocessed NOE peak lists »?

Unprocessed or raw data refers to peaks picked (or peak lists) directly from 3D or 4D NOESY data using a S/N ratio of 5 or greater. We have added a clarification for this point in the revised manuscript (pages 3 and 6).

- The server to run MAUS is functional with the data provided from the help page. However, I never got it to work with a 4D HCCH peak list in Sparky format. I do not blame the authors for that, it can certainly be that I did not provide data in the expected format. Yet, the error message obtained from the server upon failure is generic and, while it suggests how to rerun, it does not mention the actual source of error (format, incorrect sequence etc).

We apologize that 4D HCCH peak list tried by the reviewer generated a generic error message. The generic error message output by MAUS means that the data graph is not sub-graph isomorphic to the structure graph, either due to one of more NOEs which fall outside our default distance threshold of 10 Å (very unlikely), or due to incorrect inputs provided in the 2D reference file (more likely). It is also entirely possible that the user provided the data in an incorrect format. We have added this clarification in the output presented to the user and suggested means to address this issue.

- In the method section, subsection “Generating a structure graph, G”. The equation to obtain an average methyl-methyl distance uses a “r-6-averaging” (i.e. with multiplicity correction) instead of the more commonly used “r-6 summation”. Can the authors justify the use of “r-6-averaging”?

When calculating an effective distance between the 9 pairs of methyl protons, the average value of the distances (i.e., multiplicity correction) used here results in a minor increase in upper distance bounds relative to more commonly used r^6 summation. For instance, if we have 3 distance measures 4 Å, 5.5 Å and 7 Å, the regular r^6 summation results in a value of 3.9 Å whereas the average value is 4.7 Å, which remains within the range of the observed interproton distances. Due to the fact that MAUS explicitly considers different sidechain rotamers in addition to applying relatively large upper distance thresholds, the exact choice of distance averaging does not influence our results significantly (as opposed to a more precise estimate of upper bounds that is required for NMR structure determination applications). We have clarified this point on page 20 of the revised manuscript.

Reviewer #3 (Remarks to the Author):

Major revisions

The article of Nerli & De Paula et al. presents an interesting study about automated methyl group resonances assignment of protein by solution-state NMR spectroscopy. As the authors mentioned in the introduction, this is an active field of research with several recent contributions (Monneau et al. (2017), Xu et al. (2013), Chao et al. (2014), Pristisana et al. (2019)). The ultimate goal is crucial because large proteins cannot be studied with conventional NMR assignment approaches. Therefore, only methods relying on sparse NMR restraints can overcome this bottleneck. The presented approach would be to the benefit of the whole structural biology community.

The authors claim to outperform the currently available methods, which would be a significant step forward. But some important points are missing.

I strongly recommend to address these points before publication:

We thank the reviewer for their positive remarks on the importance of our work for the general, NMR-focused structural biology community. We provide a point-by-point response to the points raised by the reviewer, with revisions highlighted in yellow in the manuscript.

MAUS Robustness:

1. “MAUS uses the NOE network together with additional experimental inputs, such as peak residue type information and geminal methyl resonance connectivities, to build a system of hard constraints.”

How MAUS performs if the experimental input data contain inconsistencies?

The reviewer raises a valid point: since the different inputs to MAUS are treated as “hard constraints”, data inconsistencies can lead to either an unsatisfiable system of constraints (no output) or to incorrect assignment solutions. We would like to distinguish between two different types of possible inconsistencies, and how these are addressed by our workflow below:

- 1) Inconsistencies in peak residue type information or geminal methyl connectivities in the inputs provided by the user (such as misidentification of a Leu peak as a Val or misidentification of geminal methyls) will in most cases lead to an unsatisfiable system of constraints due to inconsistency with the global NOE network. To avoid this and guide the user towards obtaining confident assignments, in the MAUS input page we recommend that the user should only provide residue type and geminal information if they are 100% confident, guided by conclusive evidence from a range of additional experimental inputs outlined in Figure 2 of our revised manuscript. These include residue type and proS-selective labelling, 2D constant-time HMQC experiments which achieve selective inversion of the Leu peaks (Supplementary Fig. 7), further assisted by the observation of strong geminal NOEs. We find that, when provided with this information, determination of geminal methyls and residue types can be achieved with 100% confidence by the user as shown by our

results on 8 benchmark and blind targets. Alternatively, if the user is not confident in the residue type and geminal connectivities, our online interface fully supports a mixed input, where for some methyls this information is either not given or a list of options is provided (e.g. Leu or Val). Finally, the user also has the option to engage the MAUS residue type and geminal methyl classifier which provides 99% accurate results for a sub-set of methyl resonances, while leaving cases where non-confident predictions can be made as ambiguous. Together, these options effectively address any possible inconsistencies in the residue type / geminal methyl user inputs.

- 2) Inconsistencies in the input NOE peak lists, due to spectral artifacts, errors in peak picking which may also include incorrectly identified peak centers. To address this type of data inconsistency, we recommend that users pick 3D or 4D NOESY data at sufficiently high signal to noise ratio levels ($S/N > 5$). However, the most critical form of proactively removing inconsistent data points is provided by our automated clustering and symmetrization procedures, which requires that every valid NOE constraint: 1) corresponds to a single methyl group on the 2D $^{13}\text{C}/^1\text{H}$ reference HMQC spectrum, through the explicit consideration of all possible clustering scenarios and 2) arises from two exclusively symmetry-related cross peaks, through the construction of an explicit bipartite graph connecting upper- and lower-diagonal NOEs (Figure 1c and 1d). The information of which NOE peaks were not considered in subsequent graph matching steps by MAUS due to violating either point 1 or 2 is provided back to the user in a comprehensive table.

We have clarified these important points raised by the reviewer on page 25 of the revised manuscript.

2. Were the experimental peak lists manually curated / idealized? The peak lists should not be cleaned by the authors. The programs ccpnmr, sparky, or nmrDraw can automatically pick peaks. How MAUS behave with artifacts in the input peak files?

In this work, the experimental peak lists were picked manually using a S/N threshold of 5 and guided by the 2D reference spectrum. When two different users in our group picked the same data sets, they arrived at exactly the same assignment result using MAUS, suggesting that this approach is robust to any biases introduced by the user. This simple process allows us to construct NOE peak tables for each target in under 1 hr. Integrating automated pickers provided by the programs CcpNMR, sparky, or nmrDraw will result in additional time savings (and ease of use). While this is an avenue that is currently pursued in our group, it requires additional code development and extensive testing of parameters, which falls outside the scope of our current study. We have added a statement explaining this point under “*Input to MAUS*” section on page 24 in the revised manuscript.

MAUS Solver:

3. “MAUS leverages a special-purpose constraint satisfaction solver”

This is the central part of the methodology, and it is not really described. I understand that the mathematical description of the solver is beyond the scope of the publication, but a short description should be included.

We thank the reviewer for their suggestion. MAUS is using the CryptoMiniSAT satisfiability solver and a custom satisfiability formula that is specifically designed to address the methyl assignment problem. This is a well-established solver and is extremely efficient since it is aimed at solving cryptographic problems which typically tend to have much larger solution spaces. We have added a short description of the solver, and a reference to the original paper for interested readers on page 22 in the revised manuscript.

4. It is not clear if MAUS will always find all exact solutions to the NP-complete problem or if the algorithm is partially heuristic. This should be clearly stated.

We apologize that this important point did not come across in our original manuscript. MAUS always finds all possible solutions to the NP-complete problem, which is an important innovation element of our work. This is because we are casting the resonance assignment problem as a sub-graph isomorphism problem, and then further reduce it to a boolean satisfiability problem. We find that, by employing an efficient satisfiability solver, CryptoMiniSAT (which can verify if a given solution is valid or not in millisecond time scale) we can explore the space of all solutions for both simulated and real-life cases in a matter of minutes on a single CPU (see also our response to the following question No. 5). We have clarified this point on pages 21 and 22 of the revised manuscript.

5. Explain how MAUS can find the exact solutions of an NP-complete problem in a short time. To me, either MAUS is partially heuristic or finding the solution should be “slow”. Please explain that point in the manuscript.

We thank the reviewer for bringing up this important point that is a key innovative element of our method. MAUS casts the methyl assignment subgraph isomorphism problem as a system of Boolean constraints, that is described by a satisfiability formula, and uses CryptoMiniSAT, a general-purpose solver which provides either *i*) a single graph matching solution at a milliseconds-timescale on a single CPU, or *ii*) a mathematical proof that the formula is unsatisfiable. Due to the fact that it uses a custom-built satisfiability formula, MAUS has the flexibility to add or remove constraints from the formula and perform 1000s of calculations of the subgraph isomorphism problem in a feasible time frame. Using these two principles of 1) employing a custom-based formula encoding our system of hard constraints and 2) leveraging an efficient fast solver, MAUS can perform an exhaustive enumeration of the space of solutions, despite the NP-completeness of the problem, using the following iterative ansatz:

- (i) Obtain a single valid mapping of peaks into methyls (validity is imposed by the hard constraints) from the solver
- (ii) Starting from this valid mapping, consider an arbitrary peak (p1) that has been assigned to a particular methyl (m1) in (i), and add a temporary clause to the propositional formula that p1 cannot be assigned to m1
- (iii) Invoke the satisfiability solver a second time to see if it can return any alternative solution for that peak. If it can, then the propositional formula is modified again to include the new solution to the list of excluded assignments, and this process is repeated until the solver reaches unsatisfiability (*i.e.* there are no more assignments of p1 that can lead to a valid solution)
- (iv) Add hard constraints that p1 can only be assigned to exactly one of the possible options identified up to this point, and repeat steps (i) to (iii) to examine options for a second peak, chosen at random
- (v) Repeat step (iv) until all peaks have been considered

Through this iterative process, we can achieve full enumeration of the peak support sets (containing all valid methyl assignments) in a time that is proportional to the sum of the support set sizes, without resorting to the use of heuristics. We have added extra clarifying remarks to explain this process under the section “*Reducing subgraph isomorphism to satisfiability*” on pages 22 and 23 of the revised manuscript.

Other comments:

6. “performance of MAUS on deriving de novo, blind assignments

The assignment cannot be de novo blind if the authors know the assignments.

We refer to the Cas9 domain assignments as de novo/blind since we did not seek to obtain conventional resonance assignments for these targets prior to running MAUS. Instead, we first utilized MAUS to derive methyl resonance assignments for these targets *de novo* and then carried out a conventional assignment process, based on backbone triple-resonance experiments, supplemented by backbone-to-methyl and methyl-to-methyl NOEs.

7. “Our results using a benchmark set with known assignments and several blind targets show that, unlike previous methods, MAUS provides 100% accurate solutions thereby greatly reducing manual effort, costs and errors introduced due to human pre-processing of the data.”

The program MAGMA also provided 100% accurate solutions; please rephrase and or comment.

Through this statement we want to emphasize that, in addition to MAUS providing 100% accurate solutions, it also alleviates the need for the user to pre-process the raw NOE peaks, which may introduce additional biases in the assignment process. Specifically, MAGMA requires an explicit data graph to be constructed by the user through manual processing of 3D or 4D NOESY peaks. Therefore, MAUS also avoids erroneous edges being introduced in the data graph by the user. This point has been clarified on page 2 of the manuscript.

8. “Other publicly available methyl assignment methods (MAGIC/FLAMEnGo2.4/MAP-XSII) operating on the same inputs had a significant error rate of up to (34/16/56%).”

Please also compare to methylFLYA. methylFLYA outperforms MAGIC/FLAMEnGo2.4/MAP-XSII.

The reason we did not attempt a side-by-side comparison on our benchmark set of 8 targets in the original version of our paper is that, as opposed to MAUS that is readily available to the NMR community, MethylFLYA must be obtained via a commercial software license and requires a working knowledge of the INCLAN programming language by the user. Following the reviewer’s suggestion, we have purchased a license from the authors and carried out a detailed comparison of MAUS with MethylFLYA on the eight benchmark targets used in our study, by providing the identical inputs to run both methods. Upon comparison of the number of resonances assigned uniquely by MAUS and with high confidence by MethylFLYA, we find that MAUS outperforms MethylFLYA for 6/8 targets in our set (H β_2 m, MBP,

IL-2, HNH, REC2 and REC3). Specifically, (i) for IL-2 and REC2, MethylFLYA reports a significantly smaller number of confident assignments (38% and 10%) relative to MAUS (74% and 60%), which includes 2 resonances that are incorrectly assigned by MethylFLYA in each target, (ii) for HNH and MBP, it produces confident assignments at completeness levels that are 6 and 7% lower relative to MAUS, respectively and (iii) for H β 2m and REC3, MethylFLYA crashes (segmentation fault), producing no results. While MethylFLYA assigns a higher % (11 & 17%) of methyl resonances for two targets in our set (HLA-A01 & HLA-A02), it takes a prohibitively longer computation time (11 & 17 hr vs 15 & 17 min for MAUS) on a single CPU, and in general MethylFLYA is 2-3 orders of magnitude less efficient than MAUS (Supplementary Table 6). Therefore, in order for users to obtain results using MethylFLYA in a reasonable time they must also have access to a local CPU cluster, further limiting broad adoption of the method, while MAUS can be accessed by users through an online web-interface. Finally, it is worth noting that, while methylFLYA additionally provides low-confidence assignments for some resonances, MAUS provides a list of options for all methyl resonances which is guaranteed to contain the correct assignment, due to the fact that it is using a satisfiability-based enumeration process that is both exact and exhaustive. This comparison has been included in pages 5 and 27 of the revised text and Supplementary Tables 5 and 6.

Supplementary Table 6: Comparison of MAUS and MethylFLYA on our benchmark set of 8 targets

Target	# of methyls	# of unique assignments%		Time (in minutes)	
		MAUS	MethylFLYA	MAUS	MethylFLYA
H β 2m (AILV)	35	31/0	N/D*	7	N/D*
HLA-A01 (AILV)	87	56/0	64/0	15	704
HLA-A02 (AILV)	94	66/0	77/0	17	1024
MBP (MILVproS)	76	57/0	52/0	19	853
IL-2 (ILV)	61	45/0	23/2	5	2553
HNH (ILV)	53	47/0	44/0	4	814
REC2 (ILV)	69	41/0	7/2	5	1653
REC3 (ILV)	85	57/0	N/D*	14	N/D*

N/D* - (Not determined) MethylFLYA returned an error

%- # of methyls with 1 option correct / # of methyls wrongly assigned

9. “To circumvent the need for existing backbone assignments, methyl NOE data and a known structure of the target protein are used to derive a set of possible assignments, by fitting local networks of NOEs to methyl distances derived from the 3D structure...”

The mentioned methods also use long-range NOEs. Explain why you call it “local” or modify the statement.

The reviewer is correct that the mentioned methods use long-range NOEs. The “local” definition used in our manuscript refers to the approach used to fit the NOE networks to distances measured in the structure. In particular, these methods start from fitting sub-sets of NOE connectivities to local clusters of methyls in the structure and then expand those to derive self-consistent assignments for the remaining methyl resonances. Here, the NOE peak intensities are also used in an optimization process aiming to further reduce the space of solutions. This is in contrast to the global graph matching strategies used by both MAGMA and MAUS, which does not rely on NOE signal intensities that can lead to additional sources of error when used to establishing hard distance bounds, as outlined in detail in our manuscript. To avoid the confusion between “local” NOEs and networks, we have added a clarifying statement on page 2 of the revised manuscript.

10. “exhaustive mapping of the global NOE network to the target structure. This strategy was first outlined in the method MAGMA” ... “enumerate all assignments which satisfy the maximum number of NOE connectivities.”

Similar approaches were employed in a different context. I suggest the authors to read the article of Constantine et al. as well as Orts et al. (see below).

The first reference also covers bipartite graph matching problem, while the second uses sparse NOE data and also enumerates all assignments compatible with the NOE data set.

K. L. Constantine, M. E. Davis, W. J. Metzler, L. Mueller and B. L. Clau (2006) J Am Chem Soc 128 22 7252-63
Protein-ligand NOE matching: a high-throughput method for binding pose evaluation that does not require protein NMR resonance assignments

J. Orts, M. A. Walti, M. Marsh, L. Vera, A. D. Gossert, P. Guntert and R. Riek (2016) Journal of the American Chemical Society 138 13 4393-4400

NMR-Based Determination of the 3D Structure of the Ligand-Protein Interaction Site without Protein Resonance Assignment

We thank the reviewer for pointing out these important prior articles in the field. We acknowledge that bipartite graph matching methods have been previously described to address many problems in computational chemistry, including the NMR assignment problem (as described by Constantine et al.). Since the time complexity of standard matching algorithms is of the order of N^3 , finding maximum matchings in a bipartite graph that has a large number of nodes can quickly become intractable. The NMR² method by Orts et. al., relies on assigned resonances of ligand NOEs, and unassigned intermolecular NOE distance restraints together with a known protein structure to guide the search for optimal ligand interaction sites (or binding pockets), thereby eliminating the resonance assignment step for the protein component of a protein-ligand complex. However, the lack of established protein resonance assignments makes the problem of determining structures of ligand binding sites using ambiguous intermolecular NOEs extremely challenging to be solved efficiently using a global search approach. As highlighted in the manuscript, for a protein case with 54 methyls and 10 intermolecular restraints, the combinations of possible NOEs between assigned ligand and unassigned protein is approx. 10^{17} . Therefore, NMR² reduces this space by establishing hard constraints using bounds smoothing. In addition, it starts with a subset of selected distance restraints and incorporates new restraints gradually into the structure calculation procedure. The use of this strategy of applying restraints incrementally, starting from a local optimum sub-set, makes NMR² effectively a local search method (thereby not exact and non-exhaustive), as opposed to the global approach employed by MAUS' satisfiability solver leading to the search being both exact and exhaustive. As pointed out in our manuscript, one novel element of our approach relative to existing methods is that it constructs a bipartite graph to identify pairs of upper- and lower-diagonal peaks that are exclusively symmetry-related, which allows us to consider all possible NOE connectivities which can be derived from the raw NOE peaks without any intervention by the user. Another novel element is our strategy to identify all possible maximum matchings in sub-graphs of smaller sizes (what we refer to as complex components) which allows us to use an $O(N^3)$ algorithm efficiently. Finally, the bipartite graph matching process is iterated many times as an "outer loop", providing possible constraints to our satisfiability solver, which serves to eliminate some of the possible matchings. This allows us to identify a maximal, self-consistent set of upper/lower diagonal NOEs, which is used to provide the final constraints to the satisfiability solver, completely removing this burden from the user. To the best of our knowledge, this specific use of a bipartite graph matching approach has not been previously described in the NMR literature. These important points have been added under "*Reducing complex components of the symmetrization graph*" subsection on page 23 of revised text.

11. Figure S1.

Define the "effective degree connectivity"

We have added a definition of the effective degree connectivity (calculated as $2 \times$ no. of edges of the data graph/ no. of nodes) in Supplementary Figure 1 of the revised Supplementary Information document.

12, 13. Table S1. Please double-check this table. The % of unique assignment are the same between different methods and the time does not support your conclusion (MAGMA is faster than MAUS).

Since both MAGMA and MAUS perform a full enumeration of the space of valid solutions to the sub-graph isomorphism problem, when provided with the same input structure and data graphs they also produce the same outputs, and as a result the % of unique assignments should be identical between the two methods. However, since MAGMA requires a significant pre-processing of the raw NOE peaks by the user to construct the input data graph, which can also introduce biases, we did not attempt to run MAGMA on the 8 benchmark targets with new data measured in our paper. Instead, to compare our method with MAGMA we provided both methods with the same inputs (isomorphic structure and data graphs derived from the MAGMA benchmark), and the results are reported in Supplementary Table 1. We find that, while MAGMA is marginally faster than MAUS for targets that can be solved relatively quickly by both methods (in under 10s), the performance of MAUS is consistent across all targets of different sizes, including larger, more complex targets such as MSG and EIN1 for which MAGMA times out. Following the reviewer's suggestion, we have revised the manuscript on page 2 accordingly with our conclusion that MAUS has a more robust performance for larger targets, relative to MAGMA.

14. Please describe Table S5.

The Supplementary Table 5 (now Supplementary Table 4) describes the results using a residue type classifier for Leu/Val, which means that the residue types of Leucine and Valine methyl peaks in the 2D input list were replaced with the ambiguous LV annotation, and an automated classifier within MAUS was used to provide 99% confident residue type classifications for some peaks, based on their chemical shifts. Instead, LV peaks which could not be confidently classified by our method, remained ambiguous during the subsequent MAUS resonance assignment enumeration steps. To invoke the residue type classifier, the user can select "Auto" for the "Peak residue type assignment" field in the input form. This description was added to the Supplementary Table 4 in the revised Supplementary Information document.

REVIEWERS' COMMENTS

Reviewer #1 (Remarks to the Author):

The authors have completed a thorough revision of the manuscript, clarifying essential features of the method in the main text, and providing new Supplemental figures that bolster the rigor of the analysis. In addition, the authors clearly demonstrate the practical application of the MAUS tool, make reasoned counterpoints for their choice of test data, and eliminates some of the ambiguity regarding the limits of MAUS.

Based on my prior critique, I believe the manuscript can be accepted for publication.

Reviewer #2 (Remarks to the Author):

The authors have considerably improved the manuscript and most of my previous concerns have been addressed. I thus recommend publication after considering the following final comments:

1) Comparison with methylFLYA:

The results shown in Supplementary Table 6 (Comparison of MAUS and MethylFLYA on our benchmark set of 8 targets) could be added to Table 2 (Performance comparison of MAUS, MAGIC, FLAMEnGO2.4 and MAP-XSII).

The fact that methylFLYA has a commercial license cannot be a reason to exclude it from the main comparison results.

2) Question on distance threshold:

It is still unclear if the presented strategy for finding the optimal threshold is automatically applied within the MAUS protocol for the presented benchmark and is also implemented on the webserver. This should be made more clear.

Reviewer #3 (Remarks to the Author):

The authors have addressed all the issues raised. Overall, the quality of the article has significantly improved.

Reviewer #1 (Remarks to the Author):

The authors have completed a thorough revision of the manuscript, clarifying essential features of the method in the main text, and providing new Supplemental figures that bolster the rigor of the analysis. In addition, the authors clearly demonstrate the practical application of the MAUS tool, make reasoned counterpoints for their choice of test data, and eliminates some of the ambiguity regarding the limits of MAUS.

Based on my prior critique, I believe the manuscript can be accepted for publication.

We thank the reviewer for their constructive feedback that has helped us improve the manuscript significantly.

Reviewer #2 (Remarks to the Author):

The authors have considerably improved the manuscript and most of my previous concerns have been addressed. I thus recommend publication after considering the following final comments:

We thank the reviewer for their suggestions and comments. We have addressed the comments made by the reviewer in our point-by-point response below.

1) Comparison with methylFLYA:

The results shown in Supplementary Table 6 (Comparison of MAUS and MethylFLYA on our benchmark set of 8 targets) could be added to Table 2 (Performance comparison of MAUS, MAGIC, FLAMEnGO2.4 and MAP-XSII).

The fact that methylFLYA has a commercial license cannot be a reason to exclude it from the main comparison results.

We agree with the reviewer's suggestion and we have moved the results shown in Supplementary Table 6 to Table 2 of the revised manuscript, including methylFLYA in the comparison.

2) Question on distance threshold:

It is still unclear if the presented strategy for finding the optimal threshold is automatically applied within the MAUS protocol for the presented benchmark and is also implemented on the webserver. This should be made more clear.

The strategy of finding the optimal distance threshold is carried out manually by the user, according to the procedure outlined in the Methods section of our revised manuscript. To carry out this step, the user performs a set of short preliminary MAUS runs using our online web-server. This process was also performed for all targets in our benchmark set. We have added a clarifying statement on sub-section "Inputs to MAUS" (p 17). Moreover, we now provide explicit instructions on how users can optimize the NOE distance threshold in the help page of the MAUS web-server .

Reviewer #3 (Remarks to the Author):

The authors have addressed all the issues raised. Overall, the quality of the article has significantly improved.

We thank the reviewer for their constructive criticism and positive appraisal of our work.